# Photothermal and Reorientational Contributions to the Photomechanical Response of DR1 Azo Dye-Doped PMMA Fibers

**Zoya Ghorbanishiadeh** (ID), **Bojun Zhou** (ID), **Morteza Sheibani Karkhaneh** (ID), **Rebecca Oehler** (ID) and **Mark G. Kuzyk** *(ID)

Department of Physics and Astronomy, Washington State University, Pullman, WA 99164, USA; z.ghorbanishiadeh@wsu.edu (Z.G.); bojun.zhou@wsu.edu (B.Z.); morteza.sheibani.2016@gmail.com (M.S.K.); becka.oehler@wsu.edu (R.O.)
* Correspondence: kuz@wsu.edu

**Abstract:** This work is a comprehensive experimental and theoretical study aimed at understanding the photothermal and molecular shape-change contributions to the photomechanical effect of polymers doped with azo dyes. Our prototypical system is the azobenzene dye Disperse Red 1 (DR1) doped into poly (methyl methacrylate) (PMMA) polymer formed into optical fibers. We start by determining the thermo-mechanical properties of the materials with a temperature-dependent stress measurement. The material parameters, so determined, are used in a photothermal heating model—with no adjustable parameters—to predict its contribution. The photothermal heating model predicts the observations, ruling out mechanisms originating in light-induced shape changes of the dopant molecules. The photomechanical tensor response along the two principle axes in the uniaxial approximation is measured and compared with another independent theory of photothermal heating and angular hole burning/reorientation. Again, the results are consistent only with a purely thermal response, showing that effects due to light-induced shape changes of the azo dyes are negligible. The measurements are repeated as a function of polymer chain length and the photomechanical efficiencies determined. We find the results to be mostly chain-length independent.

**Keywords:** photomechanics; dye-doped polymers; polymer fibers; photo-isomerization; photothermal heating; molecular reorientation; angular hole burning; azobenzene; stress; strain

## 1. Introduction

The control of a material's mechanical properties with light, called the photomechanical effect, is becoming the focus of intense research due to its promise in directly converting light to work [1]. The origins of the idea can be traced to Alexander Graham Bell's 19th century work, who used photomechanical materials that convert light into sound to demonstrate a photophone [2,3]. Photomechanics saw a rebirth a century later when Uchino used ceramic materials to make small light-actuated "walkers" [4–7].

In addition to the promise of converting light directly into work—for example, to make motors [8–14]—materials that combine logic and actuation can underpin ultra-smart morphing technologies that leapfrog present-day capabilities [15–17]. As such, many investigations are focused on finding more efficient materials by studying the underlying mechanisms with the goal of using the fundamental knowledge that results to engineer better materials [18–23]. The observation that azo dyes can fuel a photomechanical response [24] has made dye-doped polymers an attractive material class for applications.

Harvey and Terentjev's classic experiments showed that photoisomerization of azo dye-doped elastomers is responsible for the photomechanical response [25], which has since been used to power such effects in more sophisticated materials [26]. All of these effects can be understood in models developed in the early 1990s [27]. Our work asks the question as to whether photoisomerization fuels the photomechanical response in dye-doped polymers.

The answer is sought by determining the heating contribution, using parameters extracted from temperature-dependent stress measurements that are used as an input to our model of the photothermal heating mechanism. We apply an independent measurement of the tensor components of the photomechanical response, which differentiates between heating and photoisomerization-induced angular hole burning/reorientation. In addition, the measurements are repeated as a function of chain transfer agent (CTA) concentration during polymerization, which controls the length of the polymer chain to determine if chain length matters.

The mechanical, physical and optical properties of polymeric materials depend on their molecular weight, which changes with chain length [28,29]. Thus, the ability to control the chain length of the polymer in a polymer optical fiber (POF) is critical to its performance as a waveguide and its durability in the field [29]. We focus our thermo-mechanical/photomechanical studies on azo dye-doped polymers that are drawn into fibers to study these effects in the geometry used in POF technologies, using poly (methyl methacrylate) (PMMA) as the host polymer and the azo dye Disperse Red #1 (DR1) as a representative material commonly used in POFs [17].

Chain transfer agents (CTA) limit the chain length with increased concentration as quantified by the average molecular weight of the polymer [28–31]. The best class of chain transfer agents for making PMMA by polymerizing methyl methacrylate (MMA) monomer are the thiols [28,29,31,32], due to the weakness of the S–H bond [28,29,31,33,34] and the high reactivity of the thiyl radicals [28,29,31,35–39].

The CTA molecule halts the growth of the polymer radical during polymerization [32,40,41], decreasing the molecular weight of the resulting polymer with increased CTA concentration. Many studies have shown the effect of chain transfer agent on methyl methacrylate-based polymers. Cetinkaya et al. investigated the molecular weight and optical properties of PMMA samples using different types of CTA, including isopropyl alcohol(IPA), n-butyl mercaptan (n-BMC), and pentamethyldisilane (PMDS) [42]. They showed that an increase of the CTA concentration decreased the molecular weight of the resulting bulk PMMA material [42] The optical transmittance was observed to not be affected with varying CTA concentrations, nor was the refractive index [42]. However, the refractive index varied with the different types of CTAs [42]. Moreover, the degree of scattering changed with CTA concentration only when using n-BMC [42]. They concluded that the best CTA to produce high-quality optical fiber is PMDS [42]. Valdebenito et al. studied the effect of adding 4-substituted thiophenols as CTA on the polymerization of MMA in organic media and acrylamde in aqueous solutions [28]. In the case of MMA, they found that adding thiophenols reduced the molecular weight of the polymer but did not change the polymerization rate [28].

Our work is unique in that it develops and applies models of both photothermal heating and the polarization-dependence of the photomechanical response to determine the contributions of dye shape changes and heating. This response of glassy dye-doped polymers in the form of optical fibers have not been previously characterized, nor has the effect of varying the polymer chain length on photomechanical response been studied. Our experiment is similar to others [25], but is applied in a unique way to determine other important parameters, such as the temperature-dependent stress, which is used in the models to interpret the results.

The goal of this work is to study the contributions of photothermal heating and molecular hole burning/reorientation to the photomechanical response in a glassy dye-doped polymer. photoisomerization of azobenzene molecules is well known to take place within a glassy polymer. Our work focuses on determining if these changes in molecular conformation interact with the polymer to yield a bulk mechanical response. Our approach is to couple theory and experiment to answer this question. We develop a semi-empirical theory of the photothermal mechanism, using the thermo-mechanical properties of DR1-doped PMMA fibers as a function of the CTA concentration as an input to the model, to determine its absolute contribution to the material's photomechanical constants in our experimental configuration. In parallel, we apply a theory of how the mechanisms

depend on the tensor nature of the photomechanical response, and use the consistency between these two independent experiments to pin down the contributions of the dominant mechanisms. We also report on how the polymer's chain length affects the mechanisms. The results are also used to determine the photomechanical efficiency in the fiber geometry as a function of chain length.

## 2. Theory

### 2.1. Thermal Heating

This section develops a model of the photothermal heating's contribution to the photomechanical stress response, which depends on two parameters; the stress change, $\delta\sigma$, when the temperature is increased by $\delta T$ and the response time of the photomechanical response. These parameters depend on the thermo-mechanical properties of the material and its geometry. $\delta\sigma/\delta T$ is measured in the fiber sample by monitoring the force required to keep its length fixed as a function of temperature in what we call the clamped boundary condition. The response time of the photomechanical effect is determined from the photomechanical experiment. As a first iteration, we estimate the photothermal contribution by assuming that the measured response time of the fiber is due to heating. The observation of only one time constant implies that only a single mechanism is at play. If the model's prediction of the photomechanical response matches the measured value, then it is highly likely that photothermal heating dominates.

The dynamics of the photothermal heating process is complex but can be numerically modeled [43]. Rather than doing so, we seek, here, to estimate the effect under the assumption that the observed dynamics is solely due to a temperature change, then to compare the magnitude of the measured response with that determined from the model. Then, as we describe in the next section, we develop a model that relates the stress response of both the photothermal heating and hole burning/reorientational mechanisms (see Section 2.2) to the tensor components of the stress response, which provides a second independent measurement.

The heat equation gives the rate of temperature change $dT/dt$

$$\frac{dT}{dt} = -\frac{1}{t_N}(T - T_a) + \frac{I_0 A}{mc}, \tag{1}$$

where $t_N$ is the Newton cooling time constant, which we will get from the data; assuming that the photothermal process is the only mechanism, $T$ is the elevated sample temperature due to heating, and $T_a$ is the ambient temperature. The last term in Equation (1) is the heating term, due to the intensity, $I_0$, absorbed by the sample from the light, wherein $A$ is the area of illumination, $m$ is the mass of the illuminated part of the sample and $c$ is the specific heat. In the derivation of Equation (1), we have used the fact that $\delta T$ is given by the energy deposited $\delta E = I_0 A \delta t$ in time interval $\delta t$, or

$$\delta T = \frac{\delta E}{mc}. \tag{2}$$

The temperature increase above ambient temperature at infinite time defined as $\Delta T = T_\infty - T_a$ is reached when $dT/dt = 0$, so Equation (1) yields

$$\Delta T \equiv (T_\infty - T_a) = \frac{I_0 A t_N}{mc}. \tag{3}$$

Using the fact that the heated mass $m = \rho v$ is related to the illuminated volume $v$ and polymer density $\rho$ [44,45], Equation (3) can be expressed as

$$\Delta T = \frac{I_0 t_N}{\rho c w}, \tag{4}$$

where the volume is given by $v = Aw$ with $w$ the sample thickness.

The linear photomechanical stress response $\Delta\sigma/I_0$ from photothermal heating is given by

$$\kappa_\sigma^{(1)} = -\frac{d\sigma}{dT} \cdot \frac{\Delta T}{I_0}. \tag{5}$$

As we will show later, the stress, as a function of temperature for a glassy polymer sample that is mounted between a fixed clamp and a stress sensor, is well approximated by the function

$$\sigma = \sigma_0 + \frac{\sigma_1}{\left(1 + \left(\frac{T}{T_0}\right)^n\right)}, \tag{6}$$

so we will use this as the working model for the materials studied here.

The heating contribution to the photomechanical response is analytically determined by evaluating $d\sigma/dT$ from Equation (6) and substituting it into Equation (5) with the help of Equation (4), yielding

$$\kappa_\sigma^{(1)} = -\left(-\frac{n(\sigma - \sigma_0)}{T\left(1 + \left(\frac{T_0}{T}\right)^n\right)}\right) \cdot \frac{t_N}{\rho c w}, \tag{7}$$

where $T$ is the sample temperature and we have used the fact that $A/v = 1/w$, with $w$ being the thickness of the fiber. With $\sigma_0$ and $T_0$ determined from a fit of Equation (6) to the data, and the other parameters determined as previously described, Equation (7) predicts the photothermal response.

We note that the temperature change given by Equation (4) is a linear function of $I_0$ when $\rho$, $w$ and $c$ are independent of temperature. Then, the photothermal response is strictly linear, as given by Equation (5).

To summarize, photothermal heating originates from light energy being converted to heat, which results in a temperature increase of the material. A temperature increase causes the material's equilibrium length to change. If the sample's ends are held in place by rigid clamps to prevent the material from changing length, a stress will result, which can be measured with a force sensor, as described in Section 3.2. A direct measure of the stress as a function of temperature, in essence, provides the thermal expansion coefficient. The time constant of the process, on the other hand, provides a measure of how much energy is absorbed. These two quantities together determine the amount of energy absorbed by the sample from a light beam and the stress generated.

### 2.2. Tensor Response of Heating and Reorientation

A common measurement configuration is a sample stretched between two clamps with a force sensor in series with one of the clamps [25]. This is the classical setup for tensile tests [46]. Then, the pump light illuminates the sample from the side, which can be polarized in the uniaxial stress direction or perpendicular to it. This configuration is ideal for detecting processes in which polarized light anisotropically changes the alignment or shape of the molecules because the stress measured is only along one axis (between the clamps)—call it $z$—for light polarized along $x$ or $z$. If light is polarized along $z$, the measurement yields the stress response along $z$, which is a measure of the $zz$ tensor element, which we call $\kappa_{//}^{(1)}$. If the light is polarized along $x$, the tensor element $zx$ is determined, which we call $\kappa_\perp^{(1)}$. The anisotropy is related to the ratio between $\kappa_{//}^{(1)}$ and $\kappa_\perp^{(1)}$ as described in Section 2.2.1.

Azo-dye chromophores, which change between two isomers upon light excitation, are an example of molecules that respond anisotroptically to light. In DR1-doped PMMA, the lower-energy trans molecules that are aligned along the light's polarization will preferentially be excited to the cis state. This depletes the trans molecules that originally were oriented along the light's polarization. This process is called angular hole burning [47–52],

so named because of the depletion of trans molecules oriented along the light's polarization. Figure 1 shows a schematic diagram of the process. The trans molecules are depicted in the upper part of Figure 1 by straight lines with the isotropic distribution on the upper right. Below is a polar plot of the orientation distribution function $G(\theta, \phi)$, which gives the probability density per unit solid angle of finding molecules oriented at angle $\theta$.

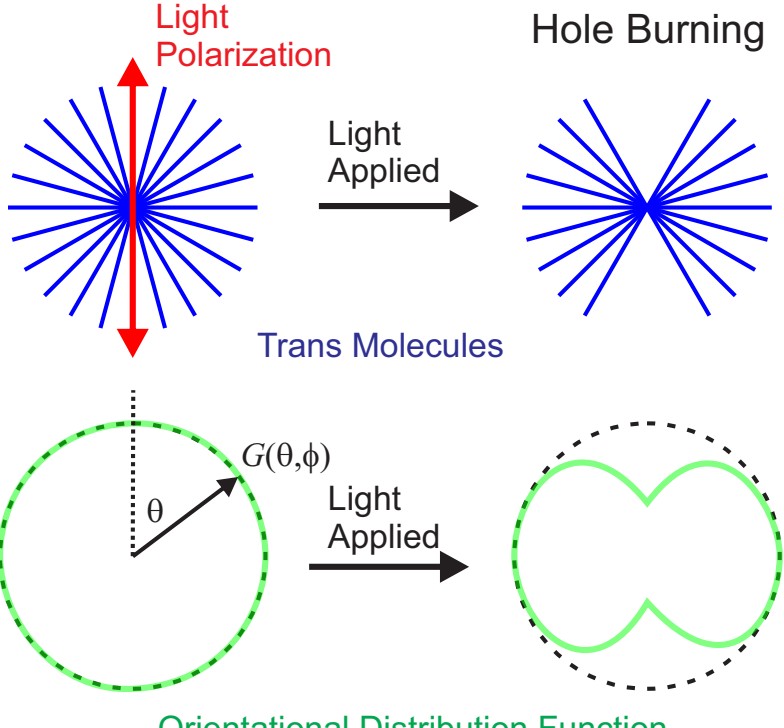

**Figure 1.** The lines in the upper plot represent an isotropic distribution of trans molecules, which are depleted along the pump light's polarization (vertical) as they are converted to cis molecules. The bottom plots show the corresponding orientational distribution functions.

Due to uniaxial symmetry, $G(\theta, \phi)$ is constant as a function of $\phi$. The depleted trans molecules appear as the pinch in the orientational distribution function. Note that the cis isomers are not shown. The area inside the polar plot of the orientational distribution function is less when the trans chromophores are depleted, a distinguishing feature of angular hole burning. Since the cis isomer is shorter, we expect that the length of the material decreases along the light's polarization and remains the same in the direction perpendicular to it. In the clamped configuration, the decrease in length induces an increase of the measured stress.

The cis isomer decays back to the trans state, but in no preferred direction. As a result, trans molecules aligned along the light's polarization vector get depleted and are subsequently isotropically redistributed. Both the excitation to the trans state and the reorientation that results when the cis molecule returns to its trans state leads to a depletion of molecules along the light's polarization and an increase of trans population perpendicular to it. This decay back to the trans state is called molecular reorientation. Figure 2 shows a schematic diagram. Note that the area of the orientational distribution function is unchanged from the original isotropic configuration. The net effect is that the stress exerted by the molecules on the polymer will decrease along the light's polarization and increase perpendicular to it as they transition to the trans state. Both angular hole burning and reorientation are at play, with angular hole burning dominating at early times and reorientation kicking in at later times. This behavior is commonly observed, with the time constants depending on the light's intensity [27,53].

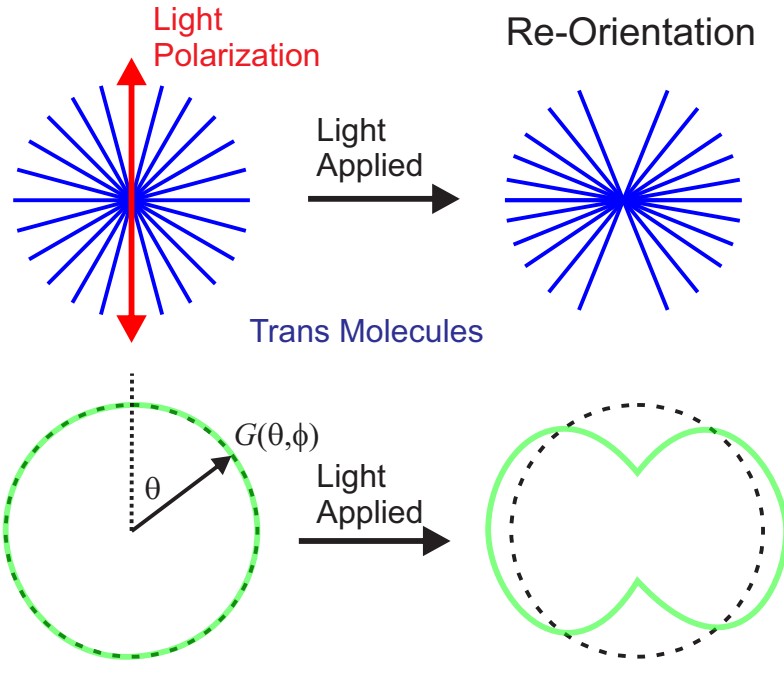

**Figure 2.** After the hole is "burned" into the material as shown in Figure 1, the cis molecules decay back into the trans state with no preferred orientation. The net effect is that trans molecules oriented along the light's polarization axis are converted into the orthogonal orientation.

If the material is initially isotropic, then photothermal heating will result in isotropic strain. Similarly, light of any polarization will deposit the same amount of heat, which will cause the material to isotropically expand. As such, the anisotropic behavior of the trans to cis photo-isomerization mechanism and the isotropic behavior of photothermal heating can be used to separate the two mechanisms in a simple experiment that records the stress response for two orthogonally polarized pump beams.

There are two cases to treat. The angular hole-burning process, shown in Figure 1, results in the photomechanical response denoted by $\kappa^{(hole)}$. The molecular reorientational process described in Figure 2 results in the photomechanical response denoted by $\kappa^{(re)}$. Below we start with a process that includes both heating and molecular reorientation. Then we treat angular hole burning.

It is possible that all three mechanisms might be acting together, making the modeling straightforward but more complex. Given the uncertainties in typical data, it is not clear weather such a model can be used to unambiguously determine the relative importance of the mechanisms. We thus focus on molecular reorientation, which builds significantly over time.

### 2.2.1. Heating and Molecular Reorientation

At low light levels, $\sigma = \sigma_0 + \kappa_\sigma^{(1)} I$, so the stress is linear in the intensity and the ratio of the stresses for the two orthogonal pump polarizations for heating and molecular reorientation is given by [50]

$$\frac{\sigma_{//} - \sigma_0}{\sigma_\perp - \sigma_0} = \frac{-\kappa_{//}^{(1)} I}{-\kappa_\perp^{(1)} I} = \frac{\kappa_{//}^{(1)}(I)}{\kappa_\perp^{(1)}(I)} = \frac{\kappa_{//}^{(heat)}(I) + \kappa_{//}^{(re)}(I)}{\kappa_\perp^{(heat)}(I) + \kappa_\perp^{(re)}(I)}, \tag{8}$$

where $\sigma_{//}$ is the stress along the sample's long axis and $\sigma_\perp$ is perpendicular to it. $\sigma_0$ is the pre-stress.

At higher intensities, where $\sigma$ is no longer a linear function of the intensity, we can define $\kappa_\sigma^{(1)}(I_0)$ to be the slope of the stress curve at intensity $I_0$, given by

$$\kappa_\sigma^{(1)}(I_0) = \left. \frac{\partial \sigma(I)}{\partial I} \right|_{I_0}. \tag{9}$$

$\kappa_\sigma^{(1)}(I_0)$ is then a material property that gives the photomechanical response for a small change in the intensity about a larger intensity $I_0$. Alternatively, we can define $\kappa_\sigma(I)$ as

$$\kappa_\sigma(I) = \frac{\sigma(I) - \sigma(0)}{I}. \tag{10}$$

Thus in the low intensity limit, the photomechanical response is linear and $\kappa_\sigma^{(1)}$ is the linear photomechanical response. $\kappa_\sigma^{(1)}(I_0)$ is the intensity-dependent linear response that quantifies the change in stress for a small change in the intensity about $I_0$, while $\kappa_\sigma(I)$ describes the change in stress for large intensity $I$ over a range where the intensity dependence is nonlinear. In the narrative that follows, we will not specify which definition is being used, but, generically, will show an intensity argument, such as $\kappa^{(heat)}(I)$. We will revisit these concepts later.

The photothermal heating contribution for an isotropic material is independent of polarization, so

$$\kappa_{//}^{(heat)}(I) = \kappa_\perp^{(heat)}(I) \equiv \kappa^{(heat)}(I). \tag{11}$$

For volume-preserving processes such as molecular reorientation, as the length of the sample decreases in one direction, it will increases in the other. If the orientational distribution function retains azimuthal symmetry, conservation of volume demands that

$$\kappa_{//}^{(re)}(I) = -2\kappa_\perp^{(re)}(I) \equiv \kappa^{(re)}(I). \tag{12}$$

Note that volume is conserved because each molecule that is depleted by reorientation is on average found to be in the plane perpendicular to the light's polarization.

Using Equations (11) and (12), Equation (8) becomes

$$\frac{\Delta\sigma_{//}(I)}{\Delta\sigma_\perp(I)} = \frac{\kappa_{//}^{(1)}(I)}{\kappa_\perp^{(1)}(I)} = \frac{\kappa^{(heat)}(I) + \kappa^{(re)}(I)}{\kappa^{(heat)}(I) - \frac{1}{2}\kappa^{(re)}(I)},$$

$$= \frac{1 + \dfrac{\kappa^{(re)}(I)}{\kappa^{(heat)}(I)}}{1 - \dfrac{1}{2}\dfrac{\kappa^{(re)}(I)}{\kappa^{(heat)}(I)}}. \tag{13}$$

Since we seek to determine the relative contribution of the two mechanisms, we define their ratio as $R^{(re)}(I)$ and invert Equation (13) to get

$$R^{(re)}(I) = \frac{\kappa^{(re)}(I)}{\kappa^{(heat)}(I)} = \frac{\dfrac{\kappa_{//}^{(1)}(I)}{\kappa_\perp^{(1)}(I)} - 1}{\dfrac{1}{2}\dfrac{\kappa_{//}^{(1)}(I)}{\kappa_\perp^{(1)}(I)} + 1}. \tag{14}$$

Equations (12) and (14) give

$$\kappa_{//}^{(re)}(I) = R^{(re)}(I)\kappa^{(heat)}(I), \tag{15}$$

and stress parallel to the light's polarization is

$$
\begin{aligned}
\kappa_{//}^{(1)}(I) &= \kappa_{//}^{(heat)}(I) + \kappa_{//}^{(re)}(I), \\
&= (1 + R^{(re)}(I))\kappa^{(heat)}(I)
\end{aligned}
\tag{16}
$$

Therefore, $\kappa^{(heat)}(I)$ and $\kappa^{(re)}(I)$ can be derived using Equations (15) and (16), yielding

$$
\kappa^{(heat)}(I) = \frac{\kappa_{//}^{(1)}(I)}{1 + R^{(re)}(I)},
\tag{17}
$$

and

$$
\kappa^{(re)}(I) = \frac{R^{(re)}\kappa_{//}^{(1)}(I)}{1 + R^{(re)}(I)}.
\tag{18}
$$

Finally, using Equations (11), (12), (17) and (18) the contribution of heating and molecular reorientation for each polarization of light is determined, yielding

$$
\kappa_{//}^{(heat)}(I) = \kappa_{\perp}^{(heat)}(I) = \kappa^{(heat)}(I) = \frac{\kappa_{//}^{(1)}(I)}{1 + R^{(re)}(I)},
\tag{19}
$$

$$
\kappa_{//}^{(re)}(I) = \frac{R^{(re)}\kappa_{//}^{(1)}(I)}{1 + R^{(re)}(I)},
\tag{20}
$$

and

$$
\kappa_{\perp}^{(re)}(I) = -\frac{1}{2}\frac{R^{(re)}\kappa_{//}^{(1)}(I)}{1 + R^{(re)}(I)}.
\tag{21}
$$

### 2.2.2. Angular Hole Burning

Angular hole burning in an azo dye such as DR1 starts with the molecule changing length when excited by light. We view the trans molecule as being uniaxial along the 3 axis as shown in Figure 3. Since the trans molecule is long and thin, we assume its orientation to be fixed by steric interactions with the polymer host. The left-hand part of Figure 3 shows such a molecule. In the cis form, the molecule becomes more compact, so we assume that it can freely reorient. Its time-averaged shape then will be a sphere, as shown by the right-hand part of Figure 3. As such, the strain along the uniaxial direction, $u_{33}$, is negative because of the decrease in its length while it is positive perpendicular to it. Figure 3 shows one molecule with its uniaxial axis tilted by an angle $\theta$ relative to the lab frame's $z$ axis.

Here we review how changes in the isomer population induced by light polarized along $z$ will change the population-averaged strain, which will lead to an anisotropic change in the material's shape. We will then use the fact that the strain is related to the stress through Young's modulus of the surrounding polymer, with which it interacts.

For a molecule oriented at Euler angles $(\theta, \phi, \psi)$, the molecular strain in the principle axis frame 123 are given by $u_{11}$, $u_{22}$ and $u_{33}$. These should be viewed as the strains that are induced by the light, which we will take into account shortly. The molecular strains are related to the strains $u_{xx}$, $u_{yy}$ and $u_{zz}$ in the lab frame through

$$
\begin{aligned}
u_{zz}(\theta, \phi, \psi) &= a_{z1}(\theta, \phi, \psi)a_{z1}(\theta, \phi, \psi)u_{11} \\
&\quad + a_{z2}(\theta, \phi, \psi)a_{z2}(\theta, \phi, \psi)u_{22} \\
&\quad + a_{z3}(\theta, \phi, \psi)a_{z3}(\theta, \phi, \psi)u_{33},
\end{aligned}
\tag{22}
$$

where $a_{z1}(\theta, \phi, \psi)$ is the $z, 1$ tensor component of the rotations matrix. A similar expression is found for $u_{xx}$ and $u_{yy}$. For a uniaxial molecule $u_{22} = u_{11}$ and if the molecules are arranged to form a uniaxial material, then $u_{yy} = u_{xx}$.

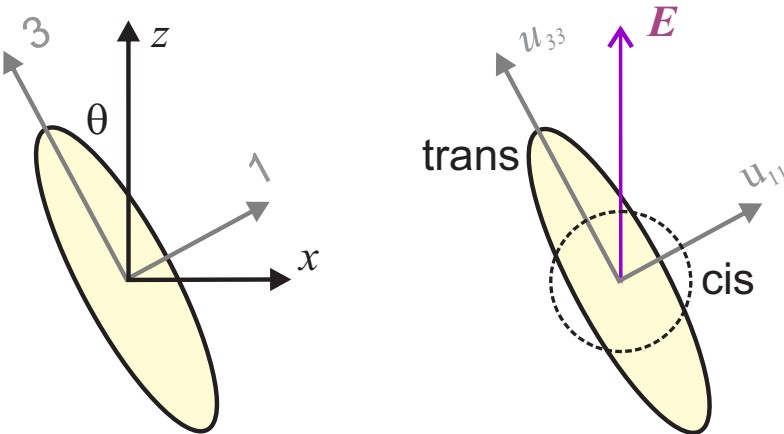

**Figure 3.** The 123 frame is attached to the molecule with 3 being the uniaxial direction and the lab frame is labelled *xyz*, where $\theta$ is the angle between the $z$ and 3 axes. The trans molecule is approximated as being a uniaxial ellipsoid and the cis isomer as a sphere. $u_{33}$ and $u_{11}$ are the strains along two of the principle axes.

A more compact way of expressing Equation (22) for all of the diagonal strain components is of the form

$$u_{ii}(\theta, \phi, \psi) = \sum_{I=1}^{3} a_{iI}(\theta, \phi, \psi) a_{iI}(\theta, \phi, \psi) u_{II}, \tag{23}$$

where $I$ is the component in the molecular frame and $i$ in the lab frame.

Light polarized along the $z$-direction in the lab frame of magnitude $E_0$ will have a component along the molecule's 3-axis of $E_0 \cos \theta$. We assume that the molecule absorbs light with an the electric field component along the 3-axis. The light-induced strain is proportional to the light energy absorbed and the absorbed energy is proportional to the square of the electric field component along the molecule's axis $E_0^2 \cos^2 \theta$. The crucial part here is that the angular dependence of the light-induced strain is given by $\cos^2 \theta$.

Putting it all together, Equation (23) becomes

$$u_{ii}(\theta, \phi, \psi) = \sum_{I=1}^{3} a_{iI}(\theta, \phi, \psi) a_{iI}(\theta, \phi, \psi) u_{II}^{(0)} \cos^2 \theta, \tag{24}$$

where $u_{II}^{(0)}$ is the light-induced strain for a molecule whose 3 axis is aligned with the $z$ axis. Equation (24) has built into it the assumption that only light polarized along the long length of the trans isomer absorbs light, vanishing when $\theta = \pi/2$.

If the material is isotropic, the molecules are initially randomly oriented, so the averaged strain matrix in the lab frame is given by

$$\bar{u}_{ii}(\theta, \phi, \psi) = \int d\Omega \, u_{ii}(\theta, \phi, \psi)$$

$$= \int d\Omega \sum_{I=1}^{3} a_{iI}(\theta, \phi, \psi) a_{iI}(\theta, \phi, \psi) u_{II}^{(0)} \cos^2 \theta, \tag{25}$$

where we have used Equation (24). $\Omega$ corresponds to the triplet of the three Euler angles and $d\Omega$ is the triple integral over the three Euler angles. Details regarding the rotation matrix and evaluating such integrals can be found in the literature [51,54].

This kind of model is similar to ones found in the literature that describe angular hole burning and molecular reorientation and which are responsible for many phenomena [14,47–50,55–60] We evaluate Equation (25) using the form of the rotation matrix given in the literature [54], which yields

$$\frac{\bar{u}_{xx}}{\bar{u}_{zz}} = \frac{4u_{11}^{(0)} + u_{33}^{(0)}}{2u_{11}^{(0)} + 3u_{33}^{(0)}} = \frac{4\frac{u_{11}^{(0)}}{u_{33}^{(0)}} + 1}{2\frac{u_{11}^{(0)}}{u_{33}^{(0)}} + 3}. \tag{26}$$

Equation (26) provides the average strain ratio, $\bar{u}_{xx}/\bar{u}_{zz}$, of the bulk material in terms of the molecular strain ratio $u_{11}^{(0)}/u_{33}^{(0)}$, induced by the light for an isotropic material.

If the microscopic strain were isotropic, or $u_{33}^{(0)} = u_{11}^{(0)}$, we would get $\bar{u}_{xx}/\bar{u}_{zz} = 1$—an isotropic bulk response, as expected. If the molecule changed length only along the $z$ direction, so that $u_{11}^{(0)}/u_{33}^{(0)} = 0$, we would get $\bar{u}_{xx}/\bar{u}_{zz} = 1/3$. In the volume-preserving case, where $u_{11}^{(0)} = -u_{33}^{(0)}/2$, we would get $\bar{u}_{xx}/\bar{u}_{zz} = -1/2$. This is the result for molecular reorientation.

Here we will treat two separate cases to get the strain.

**CASE I**

The simplest model assumes the trans molecule to be one-dimensional and the cis isomer to be point-like, as was recently shown to correctly predict the observed behavior in a pump-probe measurement [61] Then, $u_{11}^{(0)} = 0$. Substituting this into Equation (26) yields

$$\frac{\bar{u}_{xx}}{\bar{u}_{zz}} = \frac{\bar{\sigma}_{xx}}{\bar{\sigma}_{zz}} = \frac{1}{3}, \tag{27}$$

where we use the fact that the ratio of stress to strain is Young's modulus, and because the material is isotropic, so is Young's modulus, making the stress ratio and strain ratio equal to each other.

**CASE II**

Next we assume that the azobenzene molecule's length along the 3-direction is given by the tip-to-tip distance and that the width is given by twice the largest distance of an atom from the 3-axis. The cis molecule, which is more compact and assumed to be isotropically distributed, is on average a sphere of a diameter given by the tip-to-tip distance. The strain along the 3-axis would then be the change in the tip-to-tip distance upon photo-isomerization divided by the original tip-to-tip distance, whereas the strain along the 1-axis would be calculated in the same way for the distance change perpendicular to the 3-axis going from the trans molecule's width to the sphere diameter.

Using the structure of DR1 as reported in the literature, [62–64] and the analysis by Mahimwalla et al [65], we estimate the strains of the two independent tensor elements as described above and find $u_{11}^{(0)}/u_{33}^{(0)} = -2.9$. This yields

$$\frac{\bar{u}_{xx}}{\bar{u}_{zz}} = \frac{\bar{\sigma}_{xx}}{\bar{\sigma}_{zz}} \approx 3.8. \tag{28}$$

Angular hole burning needs to be added to the theory of the heating and molecular reorientation theory above to complete the model. However, this adds additional parameters that may be difficult to unravel. We describe in the experimental section how such effects can be eliminated from the time dependence of the photomechanical stress response.

*2.3. Photomechanical Efficiency*

The efficiency of a photomechanical material, which describes the fraction of the light's energy that is converted into mechanical work, is a quantity that is independent

of the measurement technique used in its determination. For the clamped stress response experiment that we use in this work, it is given by [49,50].

$$FOM = \frac{(\kappa_\sigma^{(1)})^2}{E},$$ (29)

where $\kappa_\sigma$ is the photomechanical stress response and $E$ is Young's modulus. *FOM* in Equation (29) is a material property so provides an absolute comparison between materials formed into any shape. As such, the photomechanical figure of merit can be used to study the properties of a material that affect the response as well as assessing how the various mechanisms fare.

The CTA concentration affects the molecular weight of the polymer, and hence Young's modulus, as well as how the DR1 molecule interacts with the host polymer. As such, varying the CTA concentration might affect the efficiency of the material, a hypothesis that we test experimentally.

## 3. Experiment

### 3.1. Sample Preparation

DR1-doped PMMA fibers are made with fixed DR1 concentration, but of different chain lengths, by adding different concentrations of 1-butanethiol chain transfer agent (CTA) to the MMA monomer during the polymerization process, which terminates chain growth through the transfer of radicals [66,67]. Figure 4 shows the chemical structures of MMA, PMMA, and DR1 trans and cis isomers.

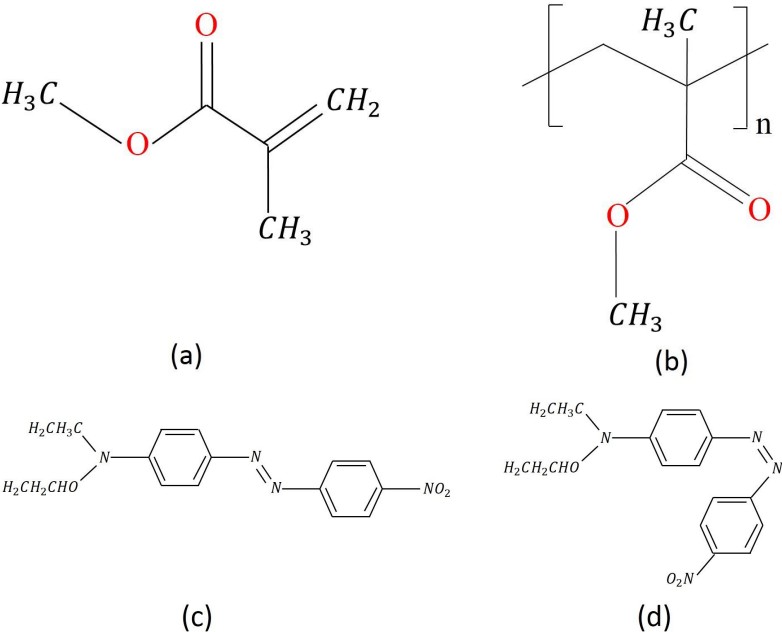

**Figure 4.** Chemical structures of (**a**) Methyl Methacrylate (MMA), (**b**) Poly (Methyl Methacrylate) (PMMA), (**c**) Disperse Red 1 (DR1) trans isomer, and (**d**) Disperse Red 1 (DR1) cis isomer.

The initiator acts as a polymerization catalyst by generating free radicals in the MMA monomer. This results in a cascade where the product radical added to the C=C bond forms a new radical upon polymerization, which can freely react with another MMA molecule to generate a polymer chain. Radical formation and chain growth propagation terminates when it encounters the chain transfer agent molecule [46,68], and thus lowers the molecular weight in proportion to the CTA concentration [42]. The lower molecular weight should lower Young's modulus [69] and $T_g$ [70] because of reduced chain entanglement [70]. Furthermore, the material's elasticity should decrease with shorter chain length.

The MMA has been purchased from Sigma Aldrich and contains $\leq 30$ ppm MEHQ (4-methoxyphenol), which acts as an inhibitor. The inhibitor has been removed from the methyl (metacrylate) (MMA) monomer solution by passing it through an alumina gravity column. DR1 has been also purchased from Sigma Aldrich with a stated purity of 95%.

Preforms are made in a test tube by adding 1-butanethiol CTA to MMA in concentrations of 20, 30, and 40 µL per 15 mL of MMA, yielding volume fractions of $1.33 \times 10^{-3}$, $2 \times 10^{-3}$ and $2.66 \times 10^{-3}$, respectively. Additionally added is 40 µL Tert-butyl peroxide initiator per 15 mL of MMA and 0.5 wt % DR1 in MMA. After polymerization, the resulting solid cylinder is machined into a rectangular shape and fibers are drawn as described in the literature [17,71].

A typical preform makes fibers of a length measured in tens of meters. Widths are 0.26 mm and thicknesses range from 0.12 mm to 0.15 mm. These are cut to the desired lengths and mounted in the sample holder, as described below. The lengths are cut to be longer than the clamp spacing to provide extra length on the ends for the grippers. The effective length of the sample is the clamp spacing, which is set to $L_0 = 8.28$ mm for both the photomechanical stress response measurement and the temperature-dependent stress measurement. The room temperature Young's modulus measurement uses a clamp separation of $L_0 = 14.05$ mm while the temperature-dependent Young's modulus measurement uses $L_0 = 14.84$ mm.

### 3.2. Photomechanical Apparatus

The sample clamp and chamber is shown in Figures 5 and 6 shows the optical path used to pump the sample. The experiment is used to measure the temperature and intensity-dependent photomechanical response of the fibers. The same apparatus is used to determine Young's modulus, as well as the stress induced by thermal expansion as the oven temperature is increased.

Figure 5 shows a fiber mounted between the clamps and the force sensor, which is attached to the adjustable upper clamp in series with the sample. The force sensor, based on an electrically resistive wheatstone bridge, converts stress changes in the sample to a voltage change, which is read by an Arduino. The brass oven chamber fully encloses the sample and allows the light to enter through glass windows, allowing the sample temperature to be programmed to change as a function of time while light is applied and stress measured. Though the stress sensor is designed to compensate for temperature changes by design, it is mounted outside of the chamber to keep temperature changes to a minimum.

The 488-nm line of a krypton/argon laser uniformly illuminates the full surface of each fiber using a cylindrical lens, as shown in Figure 6, which yields a rectangular spot whose dimensions are larger than the sample's dimensions. The light polarization is set to vertical or horizontal using the polarizer and the intensity can be varied with neutral density filters and fine-tuned with the half-wave plate. A shutter is placed before the sample to turn the light on and off with a period of 60 s. The light that is transmitted through the sample passes through a cylindrical lens that images it onto a power meter.

The experiment measures the force for a range of absorbed light intensities we call $I_0$. The stress is calculated from the force $F$ and the sample's dimensions using

$$\Delta\sigma = \frac{\Delta F}{Ww}, \tag{30}$$

where $W$ is the sample width and $w$ the thickness. From this, the photomechanical stress response $\kappa$ is determined using

$$\kappa = \frac{\Delta\sigma}{\Delta I_0}. \tag{31}$$

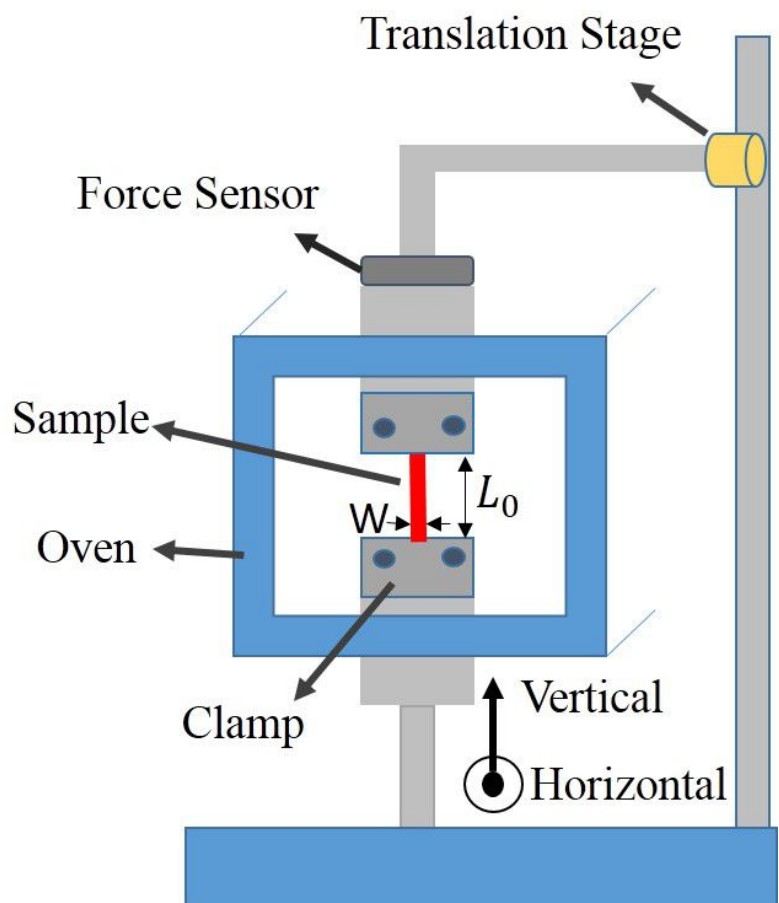

**Figure 5.** Oven chamber that contains the fiber sample and an external force sensor in series with the upper clamp used for thermo-mechanical and photomechanical experiments. The pump laser beam travels into the page and illuminates the whole face of the fiber.

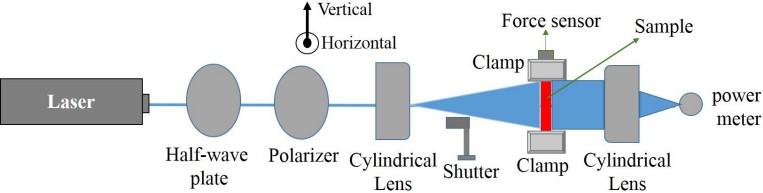

**Figure 6.** The optical path in the photomechanical experiment that is used to measure stress as a function of time, strain, pump intensity and temperature. Oven is not shown.

*3.3. Experimental Procedure*

Experiments are conducted over a temperature range from room temperature to about 320 °K—well below the glass transition to assure the sample's mechanical integrity is preserved under stress—to study the thermomechanical and photomechanical properties of DR1-doped PMMA. These experiments use three representative CTA volume fractions that yield low-, medium-, and high-molecular-weight polymer. All fibers are 0.26 mm wide and 0.12–0.15 mm thick, with their lengths determined by the clamp separation. All intensity-dependent experiments use a $\lambda = 488$ nm pump from an argon/krypton laser. The results are used to determine the material efficiency figure of merit as a function of light intensity and CTA volume fraction. The samples are pre-stressed before starting the measurements to avoid buckling.

The absorbed power is estimated by taking the difference in the power measured by a photo-detector with and without a sample in the mount. A portion of the light is reflected or scattered by the sample. Given that the dye-doped polymer's refractive index

is about 1.48, the Fresnel reflectivity is about 4% from each surface, so the absorbed power is underestimated by about 8%. Since the face of the sample is flooded with light, the ratio of absorbed power to the area of a sample's face yields the average absorbed intensity. It is this average absorbed intensity that is implied when mentioned in the text or plotted in the figures.

### 3.4. Experimental Uncertainty

All experimental uncertainties of measured parameters are determined from an estimate of the accuracy of the instrument used and, in the case of nonlinear curve fits, are determined from the fitting routines built into the Origin scientific spreadsheet software. The error is propagated for calculated quantities by the standard quadrature method from the measured parameters.

## 4. Results and Discussion

### 4.1. Temperature Dependent Measurements

Each sample is mounted between the clamps with initial length $L_0$, and Young's modulus is measured at room temperature and elevated temperatures. Figure 7 shows Young's modulus at room temperature as a function of CTA volume fraction. Young's modulus is observed to decrease with decreased average chain length.

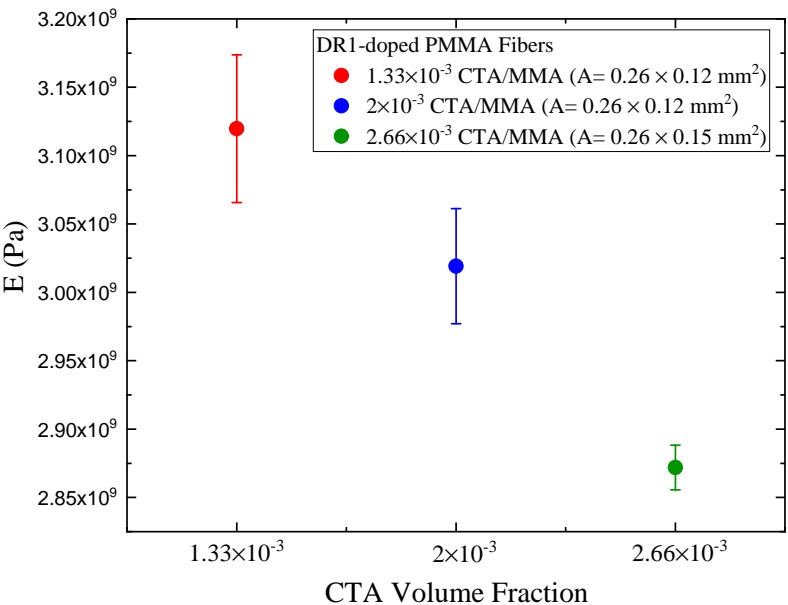

**Figure 7.** Measured Young's modulus as a function of CTA volume fraction at room temperature.

Figure 8 shows Young's modulus as a function of temperature from 293.15 °K to 323.15 °K in 3 °K increments. The samples soak for 15 min at each temperature to assure they have reached equilibrium before taking a measurement. Young's modulus is observed to decrease as a function of increasing temperature, where all samples show similar behavior. The decrease of Young's modulus with increased temperature and increased CTA volume fraction is consistent with the increased degree of chain mobility as the temperature in increased nearer to $T_g$ and the decrease of chain length with the addition of CTA during polymerization, which is associated with a decrease in $T_g$ as chain length decreases.

We propose a model for Young's modulus that takes the form

$$E = \frac{E_0}{\left(1 + \left(\frac{T}{T_c}\right)^n\right)},\tag{32}$$

where $E_0$ is Young's modulus in the $T = 0$ limit, $T$ the oven/sample temperature, $T_c$ a characteristic temperature where the Youngs modulus is half the zero-temperature value,

and *n* is a critical exponent. We chose this function because it fits the data well with a minimal number of parameters. Equation (32) provides a convenient analytical function that can be fit to the data from which one can determine material parameters, such as $dE/dT(T)$.

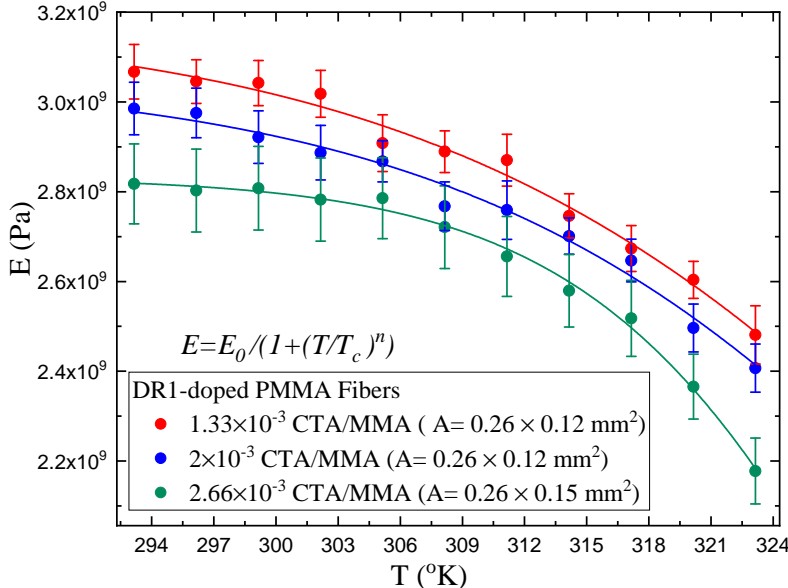

**Figure 8.** Measured Young's modulus as a function of temperature for three CTA volume fractions (points with error bars) and fits to the model given by Equation (32).

Table 1 shows the parameters obtained from fitting the data in Figure 8 to Equation (32). $E_0$ decreases as a function of CTA volume fraction, as we would expect from shorter chain length. The characteristic temperature and the critical exponent are the same within experimental uncertainties for $1.33 \times 10^{-3}$ and $2 \times 10^{-3}$ CTA volume fraction but the exponent for $2.66 \times 10^{-3}$ CTA volume fraction is much larger, as we would expect for the increased mobility of shorter chain length.

**Table 1.** Parameters determined from fitting the Young's modulus data to Equation (32) for the three CTA volume fractions.

| CTA Volume Fraction | $E_0$ (GPa) | $T_c$ (°K) | n |
|---|---|---|---|
| $1.33 \times 10^{-3}$ | 3.18 ($\pm 0.05$) | 342.5 ($\pm 2.4$) | 22.0 ($\pm 3.2$) |
| $2 \times 10^{-3}$ | 3.05 ($\pm 0.05$) | 341.6 ($\pm 2.5$) | 23.9 ($\pm 3.6$) |
| $2.66 \times 10^{-3}$ | 2.83 ($\pm 0.01$) | 332.50 ($\pm 0.60$) | 42.4 ($\pm 2.4$) |

Figure 9 shows a plot of the temperature-dependent stress of the clamped fibers for the three different chain lengths. The temperature is increased from 294 °K to 313 °K in 3 min and modeled with the function given by Equation (6) because it fits the data well and the analytical form determined from a fit to the data can be used to determine $d\sigma/dT$, which is used to model the photothermal mechanism, as described in Section 2.1. Table 2 summarizes the parameters of the stress fitting function.

**Table 2.** The parameters determined by fitting the temperature-dependent stress data in Figure 9 to Equation (6) for fibers with three different CTA volume fractions.

| CTA Volume Fraction | $\sigma_0$ (MPa) | $\sigma_1$ (MPa) | $T_0$ (°K) | n |
|---|---|---|---|---|
| $1.33 \times 10^{-3}$ | $-300$ ($\pm 0$) | 301.21 ($\pm 0.02$) | 451.9 ($\pm 2.5$) | 13.66 ($\pm 0.22$) |
| $2 \times 10^{-3}$ | $-488$ ($\pm 0$) | 488.98 ($\pm 0.01$) | 431.4 ($\pm 1.6$) | 17.49 ($\pm 0.22$) |
| $2.66 \times 10^{-3}$ | $-29.4$ ($\pm 0$) | 32.39 ($\pm 0.04$) | 374.99 ($\pm 0.33$) | 9.76 ($\pm 0.09$) |

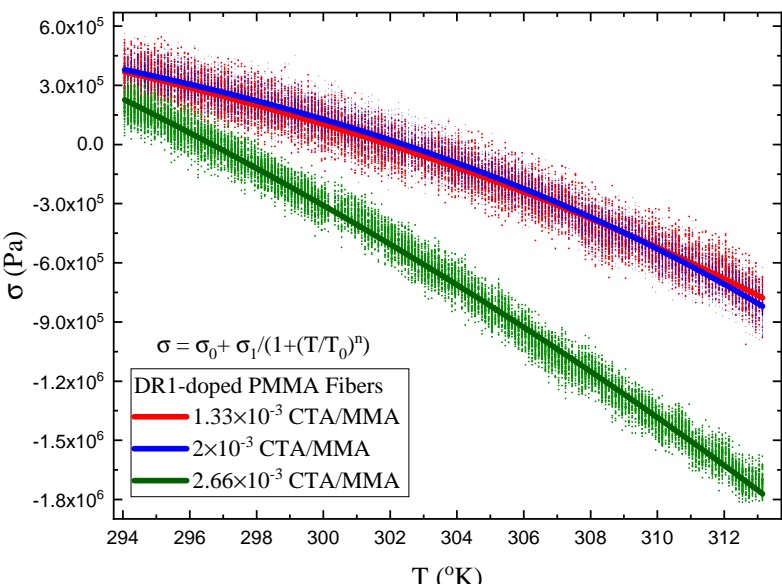

**Figure 9.** Measured stress as a function of temperature for clamped DR1-doped PMMA fibers at three different CTA volume fractions (points) and fits to the model given by Equation (6).

### 4.2. Intensity-Dependent Photomechanical Measurements

The sample is pumped with 488-nm wavelength light with a square wave intensity profile and the photomechanical stress response is measured as a function of time for three CTA volume fractions in the apparatus described in Section 3.2. The DR1 concentration is fixed in all fibers at 0.5% by weight. The stress is measured as a function of the intensity amplitude of the square wave separately for vertical and horizontal polarization of the pump light to determine the photomechanical stress response tensor. The black points in Figure 10 show typical smoothed data for a 60-s period square wave and the solid curve is a fit to a single exponential given by Equations (33) and (34) as described below.

We note that the stress in Figure 10 changes sign, so the material transitions form a stretched state to a compressed state relative to its equilibrium dimensions. As a result, buckling may take place beyond the transition at high stress. Figure 10 is representative of data for the highest pump intensities. However, most of the intensities reported here are at levels below the buckling point. Even those above the buckling point are observed to follow the trends, so buckling does not appear to be an issue for samples of this geometry. Higher pre-stress than those used is avoided to prevent sample damage or inducing anisotropies.

The data in Figure 10 behave as a single saturated exponentials in both the pump on and off regions. We require that the response in the two regions join together continuously but the time constants are permitted to be different. Putting these criteria together leads to the model

$$\sigma_{on} = \sigma_0 + \sigma_1 (1 - e^{(-t/t_1)}) \tag{33}$$

and

$$\sigma_{off} = \sigma_0 + \sigma_1 (1 - e^{(-t_0/t_1)}) e^{-(t-t_0)/t_2}, \tag{34}$$

where $A_n$ and $t_n$ are the amplitudes and time constants and $t_0 = 30\,\text{s}$—the time durations of the beam being on and the beam being off. These types of fits are commonly used [49].

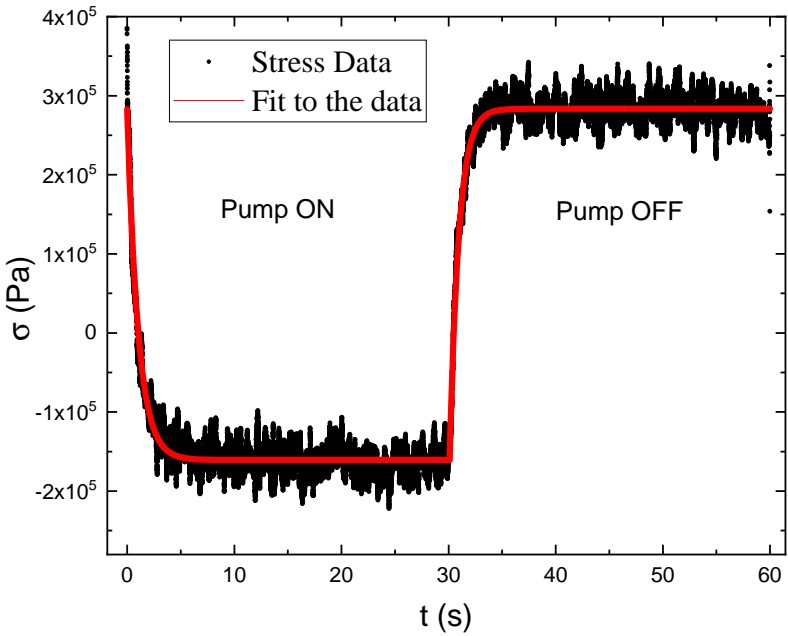

**Figure 10.** Typical time-dependent stress data (each point in the plot is smoothed over 100 points) and a fit to the theory given by Equations (33) and (34) (smooth curve) for a pump that turns on and off over a period of 60 s.

Depending on the material, light can induce an expansion or contraction. Since an increase in the length of a sample leads to a decrease in the stress for a pre-stressed sample, we define the photomechanical stress response to be of the form,

$$\sigma(I, t \to \infty) \equiv \sigma(I) = -\sum_{n=0}^{\infty} \kappa_\sigma^{(n)} I^n \approx -(\kappa_\sigma^{(1)} I + \kappa_\sigma^{(2)} I^2),\tag{35}$$

where $\sigma$ is the stress, $\kappa_\sigma^{(n)}$ the photomechanical stress response, and $I$ the intensity absorbed by the sample. With this definition, $\kappa_\sigma^{(1)}$ is positive when the length of the sample increases [49,50]. Figure 10 shows typical stress data as a function of time and fits to the theory given by Equations (33) and (34). Equation (35) assumes that the stress function $\sigma(I)$ can be expanded in a series and that the first two terms are sufficient to approximate its behavior.

Here we define the linear intensity-dependent photomechanical response, $\kappa^{(1)}(I_0)$, which quantifies the linear photomechanical response of a material that is bathed in light of intensity $I_0$. This quantifies the change in stress for a small added intensity $\delta I$ to the large background intensity $I_0$. It can be determined from a measurement of the stress $\sigma(I)$ as a function of $I$, yielding

$$\kappa_\sigma^{(1)}(I_0) = \left.\frac{\partial \sigma(I)}{\partial I}\right|_{I_0} = \kappa_\sigma^{(1)} + 2\kappa_\sigma^{(2)} I_0.\tag{36}$$

We can view the normal linear response $\kappa_\sigma^{(1)}$ as being the limiting case of $\kappa_\sigma^{(1)}(I_0)$ when $I_0 \to 0$. As such, $I_0$ provides another parameter that can be varied to test hypotheses of what makes a large photomechanical response.

The data for vertically and horizontally polarized pump beams are separately fit to Equations (33) and (34). Figure 11 shows a plot of the parameter $\sigma = \sigma_1$, the steady state light-induced stress at infinite time, which is determined from such fits at each pump

polarization. Also shown is a fit to the first two terms of the Taylor series expansion given by Equation (35). This expansion is a good approximation to the intensity-dependent response with a small quadratic contribution.

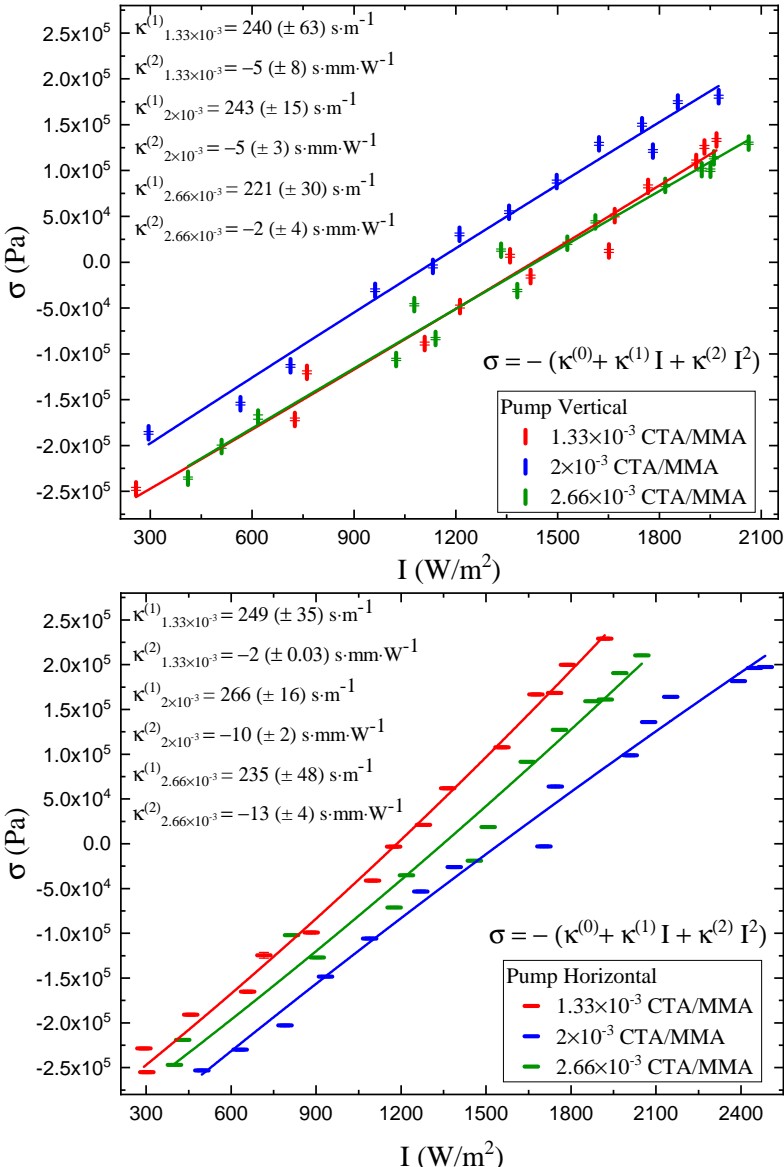

**Figure 11.** Polarization-dependent steady-state stress response at long times as a function of pump intensity at a pump wavelength of 488 nmfor different CTA volume fractions. The pump light is polarized parallel to the fiber's long axis (**top**) and perpendicular to the fiber axis (**bottom**).

The results of the linear photomechanical and nonlinear stress response $\kappa_\sigma^{(1)}$ and $\kappa_\sigma^{(2)}$ for vertical and horizontal polarization of light and as a function of CTA volume fraction are displayed in Table 3. The linear photomechanical response is the same within experimental uncertainties for all CTA volume fractions, so is independent of the polymer chain length. Furthermore, the linear response is independent of the polarization within experimental uncertainties, but the smaller values for the vertically polarized pump relative to the horizonal polarization are consistent with a small contribution from molecular reorientation.

The quadratic stress responses $\kappa_\sigma^{(2)}$ for the vertically polarized pump are all the same, within experimental uncertainty, as a function of CTA volume fraction, but the response for a perpendicular pump becomes more negative. This suggest a nonlinearity, due to a mechanism that is not isotropic, as is heating. This increase in the magnitude of the quadratic response correlates with the decrease in Young's modulus and the decrease in

polymer chain length. The chain length and Young's modulus decreases result in greater chain mobility and, hence, greater mobility of the dopant molecules, allowing them to reorient. However, the negative sign is opposite to what we would expect; with increased molecular reorientation, a horizontal pump would induce an increased stress response along the uniaxial direction.

**Table 3.** Summary of the room-temperature properties of Disperse Red 1 doped poly (methyl methacrylate) (DR1-doped PMMA) fibers as a function of CTA volume fraction, pre-stress and light polarization at a pump wavelength of 488 nm.

| CTA Volume Fraction | Polarization | Pre-Stress (MPa) | $\kappa_\sigma^{(1)}$ (s/m) | $\kappa_\sigma^{(2)}$ (s·mm/W$^2$) | E (Gpa) | FOM ($10^{-4}$ s$^2$/N ) |
|---|---|---|---|---|---|---|
| $1.33 \times 10^{-3}$ | Vertical | 0.47 | $240 \pm 63$ | $-5 \pm 8$ | $3.12 \pm 0.05$ | $0.19 \pm 0.05$ |
| $1.33 \times 10^{-3}$ | Horizontal | 0.47 | $249 \pm 35$ | $-2 \pm 0.03$ | $3.12 \pm 0.05$ | $0.20 \pm 0.03$ |
| $2 \times 10^{-3}$ | Vertical | 0.47 | $243 \pm 15$ | $-5 \pm 3$ | $3.02 \pm 0.04$ | $0.20 \pm 0.01$ |
| $2 \times 10^{-3}$ | Horizontal | 0.47 | $266 \pm 16$ | $-10 \pm 2$ | $3.02 \pm 0.04$ | $0.23 \pm 0.01$ |
| $2.66 \times 10^{-3}$ | Vertical | 0.38 | $221 \pm 30$ | $-2 \pm 4$ | $2.87 \pm 0.02$ | $0.17 \pm 0.02$ |
| $2.66 \times 10^{-3}$ | Horizontal | 0.38 | $235 \pm 48$ | $-13 \pm 4$ | $2.87 \pm 0.02$ | $0.19 \pm 0.04$ |
| $2.66 \times 10^{-3}$ | Vertical | 0.76 | $263 \pm 42$ | $6 \pm 4$ | $2.87 \pm 0.02$ | $0.24 \pm 0.08$ |
| $2.66 \times 10^{-3}$ | Horizontal | 0.76 | $266 \pm 20$ | $1.3 \pm 3$ | $2.87 \pm 0.02$ | $0.25 \pm 0.04$ |

Figure 12 shows the photomechanical response as a function of intensity calculated from Equation (36) at the measured intensities using the fit parameters $\kappa_\sigma^{(1)}$ and $\kappa_\sigma^{(2)}$ from Table 3. The intensity-dependent response for pump light polarization along the axis containing the force sensor (which we will call vertical) changes by less than or equal to approximately 10%. The heating model given by Equation (7) predicts that the change in stress induced by the photothermal heating mechanism is independent of the intensity. The data for vertically polarized pump light, thus, shows that heating is responsible for at least 90% of the observed photomechanical response.

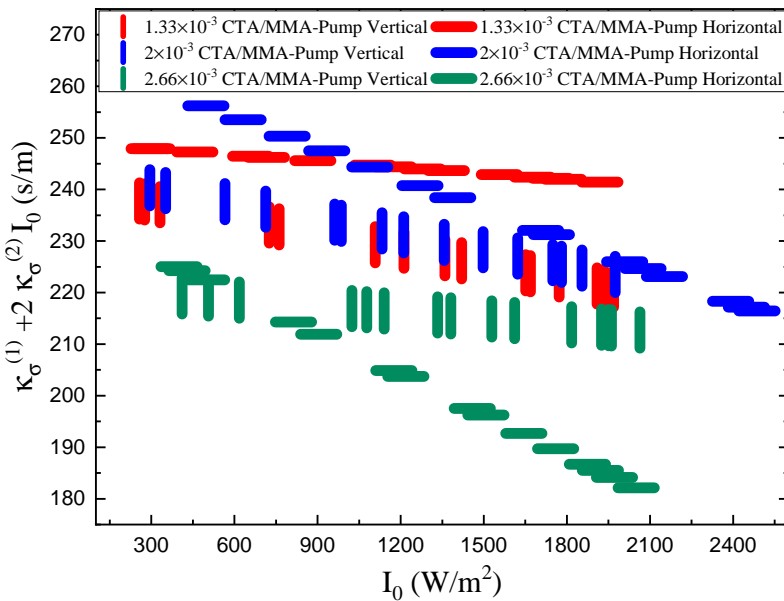

**Figure 12.** The intensity-dependent photomechanical response as a function of intensity calculated from Equation (36) at the measured intensities using the fit parameters $\kappa_\sigma^{(1)}$ and $\kappa_\sigma^{(2)}$ from Table 3.

Molecular reorientation is the process by which molecules reorient away from the pump light's polarization axis, which should lead to an increase in the measured stress as the material's equilibrium length decreases for a vertical pump polarization. Reorientation thus opposes heating for vertical polarization, which tends to decrease the stress as the

sample elongates. As a result, the response is lower than it would be with heating alone. When the light is horizontally polarized, reorientation toward the vertical direction leads to a decrease in the stress as the material's length increases, thus adding to photothermal heating, which also results in the sample elongating.

For all three polymers of varying polymer chain length, Figure 12 shows the vertically polarized pump to have a smaller response than the horizonal one in the limit of small intensity, as we would expect for a small reorientational contribution to a response that is dominated by photothermal heating. The difference between the two diagonal tensor components at room temperature is on the other of 10 s/m to 20 s/m, suggesting that the reorientational contribution is 5 s/m to 10 s/m—a 2%-to-5% contribution.

The intensity dependence is governed by the $\kappa_\sigma^{(2)} I$ term, which is more difficult to explain. It is due to both the temperature dependence of photothermal heating—which originates in the temperature dependence of material parameters such as the heat capacity, and molecular reorientation—which depends on the viscoelastic properties of the polymer and the temperature-dependence of the isomer populations. Accounting for these contributions are beyond the scope of our present work but will be the topic of future studies.

Figure 13 shows the turn-on time constants, $t_1$—when the pump is on—and Figure 14 shows the turn-off time constant, $t_2$—when the pump is off—as a function of intensity, determined from a fit of the data to the functions in Equations (33) and (34). If heating is the only mechanism, then $t_1$ and $t_2$ should be the same as the Newton cooling time constant $t_N$. The insets show the time constants as a function of CTA volume fraction averaged over all the intensities. The time constant of the polymer made with higher CTA concentrations is marginally higher than those made with lower CTA concentrations.

The Beer–Lambert law [50]

$$I = I_i e^{-\alpha w} \tag{37}$$

is used to determine the material absorbance $\alpha$ at a wavelength of 488 nm by measuring the transmitted intensity $I$ and the incident intensity $I_i$ with a photo detector. The fiber thickness through which the light propagates, $w$, is measured with a micrometer. Table 4 shows the absorbance for both incident polarizations. Note that the absorption length is much longer than the fiber thickness, so nonlinearities from depth effects can be ignored [72,73], making the assumptions in our heating model appropriate.

**Table 4.** Absorbance as a function of CTA volume fraction and light polarization at 488 nm.

| CTA Volume Fraction | Polarization | Absorbance $\alpha \, (\mathrm{m}^{-1})$ |
|---|---|---|
| $1.33 \times 10^{-3}$ | Vertical | 3.54 |
| $1.33 \times 10^{-3}$ | Horizontal | 5.59 |
| $2 \times 10^{-3}$ | Vertical | 6.85 |
| $2 \times 10^{-3}$ | Horizontal | 7.57 |
| $2.66 \times 10^{-3}$ | Vertical | 5.23 |
| $2.66 \times 10^{-3}$ | Horizontal | 6.15 |

The fibers are slightly pre-stressed when first mounted to prevent buckling when they expand in response to light or temperature change. The pre-stress may result in chain alignment and thus affect the photomechanical response. If the chains and molecules were to align along the fiber axes, the vertical absorbance would be greater than the horizontal one. We find horizontal absorbance to be higher, which is inconsistent with the hypothesis that molecular alignment is induced in the fiber drawing process and/or due to prestress.

Table 3 shows the photomechanical response, $\kappa_\sigma^{(1)}$ and $\kappa_\sigma^{(2)}$, of the fiber with $2.66 \times 10^{-3}$ CTA volume fraction for pre-stress values of 0.38 MPa and 0.76 MPa. The response is found to be the same within the large experimental uncertainties for both polarizations, though they are systematically higher for the higher pre-stress values.

The efficiency figure of merit (FOM), which is the material's efficiency in converting light energy to mechanical work, is defined in Equation (29) [49,50]. We determine the FOM as a function of CTA volume fraction. Table 3 summarizes the results. Within experimental uncertainties, the FOM is found to be independent of the CTA concentration.

These results emphasize some of the advantages of glassy polymers as host materials. They have a large Young's modulus, so they are more rigid than other materials, such as liquid crystal elastomers, and also have a larger photomechanical stress response [49,50]. Crystals, on the other hand have good photomechanical properties but have lower optical quality and cannot be formed into arbitrary shapes [74–78]. These are important qualities for providing a rigid support for the precise positioning of massive objects. Additionally, glassy polymers can be formed into thin films and fibers of high optical quality, making them promising candidates for waveguide devices and smart photomechanically morphing materials [17].

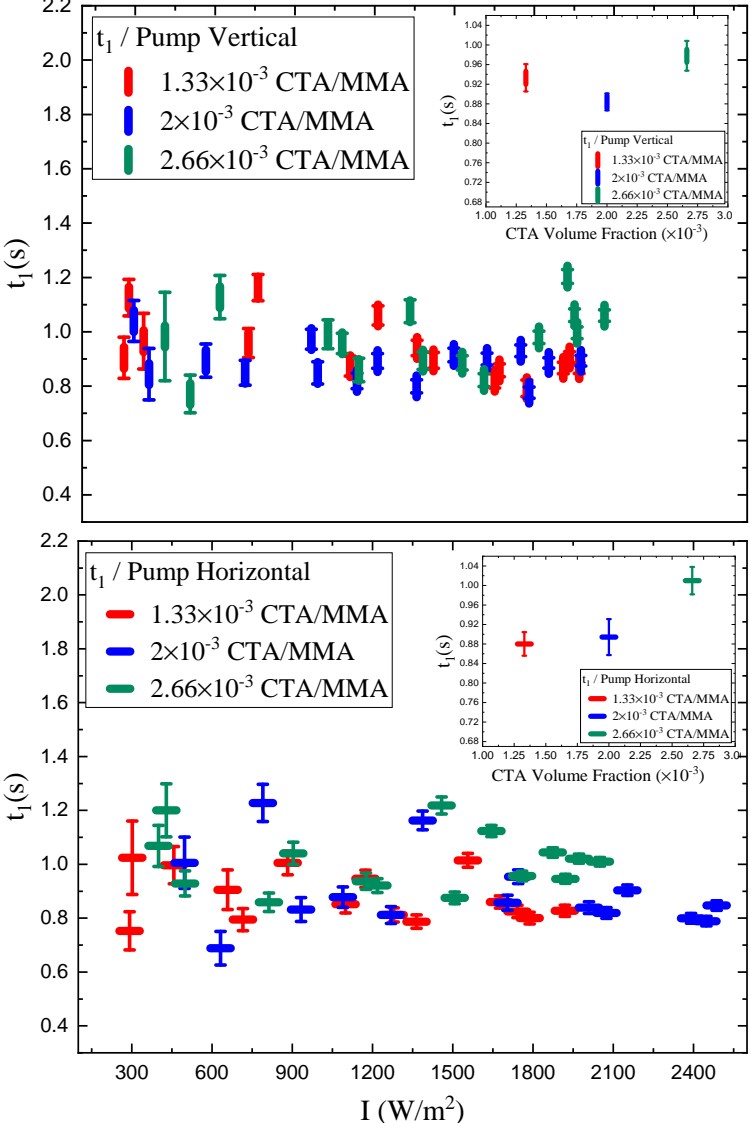

**Figure 13.** Time constant $t_1$ (light on) as a function of intensity for different CTA volume fractions for a vertically and horizontally polarized pump. The insets show the time constants as a function of CTA volume fraction averaged over all intensities.

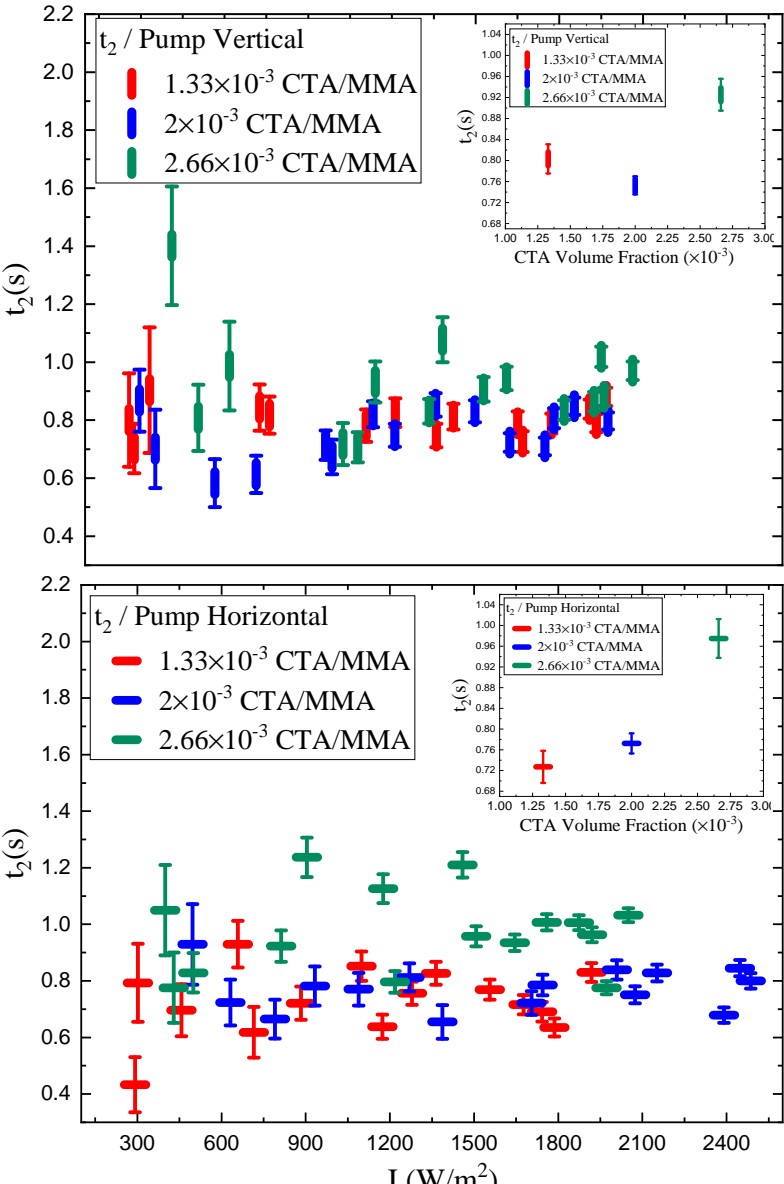

**Figure 14.** Time constant $t_2$ (light off) as a function of intensity for different CTA volume fractions for a vertically and horizontally polarized pump. The insets show the time constants as a function of CTA volume fraction averaged over all intensities.

### 4.2.1. Photothermal Heating

This section shows how the parameters in the photothermal heating model are determined from the experiment and how the tensor properties of the photomechanical response are used to isolate the heating contribution. These two independent approaches will be used to separate the photothermal response and hole burning/molecular reorientation. The degree of consistency between them will provide an estimate of the reliability of the measurements and their interpretation.

Figures 15 and 16 show the semi-empirically determined temperature change as a function of intensity using Equation (4) and the measured time constant of the photomechanical response. The theory has no adjustable parameters, and depends only on known material parameters, such as the specific heat capacity $c$ and the density $\rho$, as well as on the sample's geometry, characterized by its width, $w$. The values of $\rho$, $c$, and $w$ that are used to find the temperature increase are listed in Table 5. The insets in the graphs in Figures 15 and 16 show the temperature change per unit of absorbed intensity as a function of CTA volume fraction.

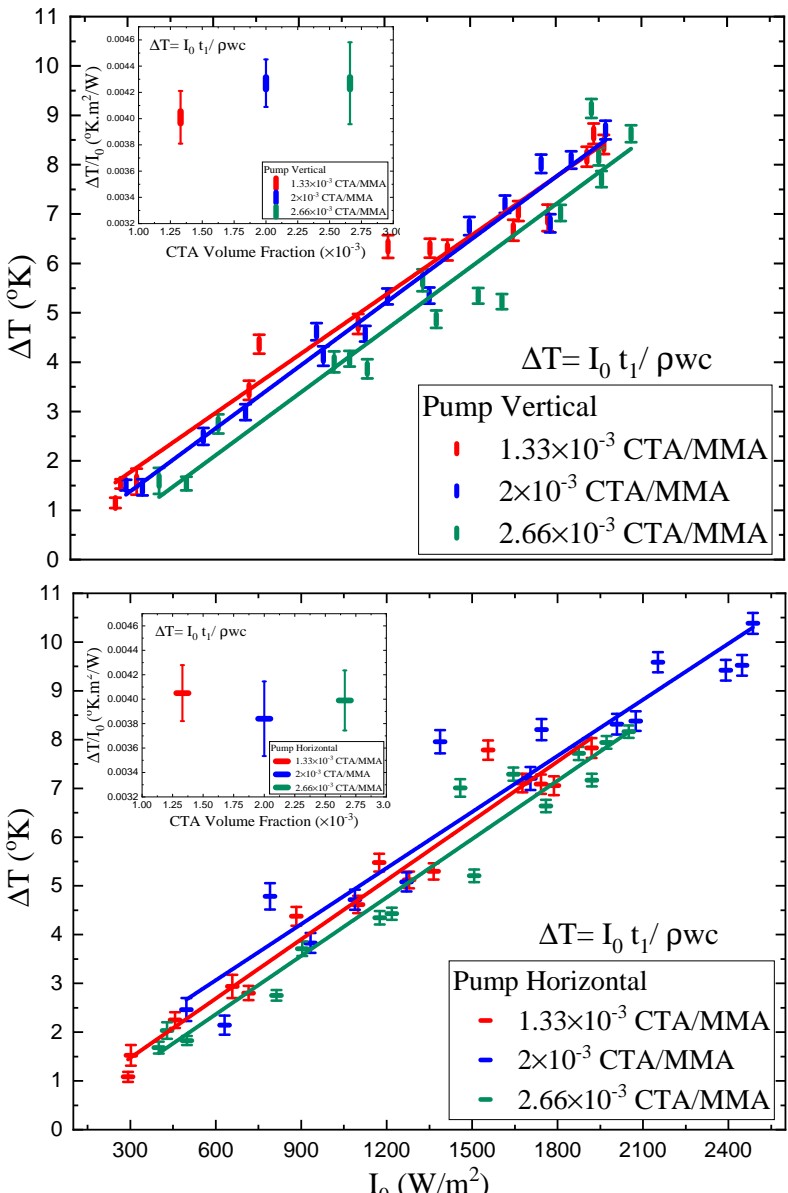

**Figure 15.** Temperature change as a function of intensity calculated from Equation (4) using the time constants measured from the photomechanical response, the density, heat capacity, and the sample's dimensions. The insets show the temperature change per unit of absorbed intensity as a function of CTA volume fraction averaged over all intensities. The plots use $t_1$ to get $\Delta T$ for vertically and horizontally polarized pumps.

**Table 5.** Values of density $\rho$, the specific heat $c$, and thickness $w$ of the fibers used to calculate the temperature increase.

| CTA Volume Fraction | $\rho$ (kg/m$^2$) | $c$ (J/kg °K) | $w$ (mm) |
|---|---|---|---|
| $1.33 \times 10^{-3}$ | $1.19 \times 10^3$ | $1.42 \times 10^3$ | 0.12 |
| $2 \times 10^{-3}$ | $1.19 \times 10^3$ | $1.42 \times 10^3$ | 0.12 |
| $2.66 \times 10^{-3}$ | $1.19 \times 10^3$ | $1.42 \times 10^3$ | 0.15 |

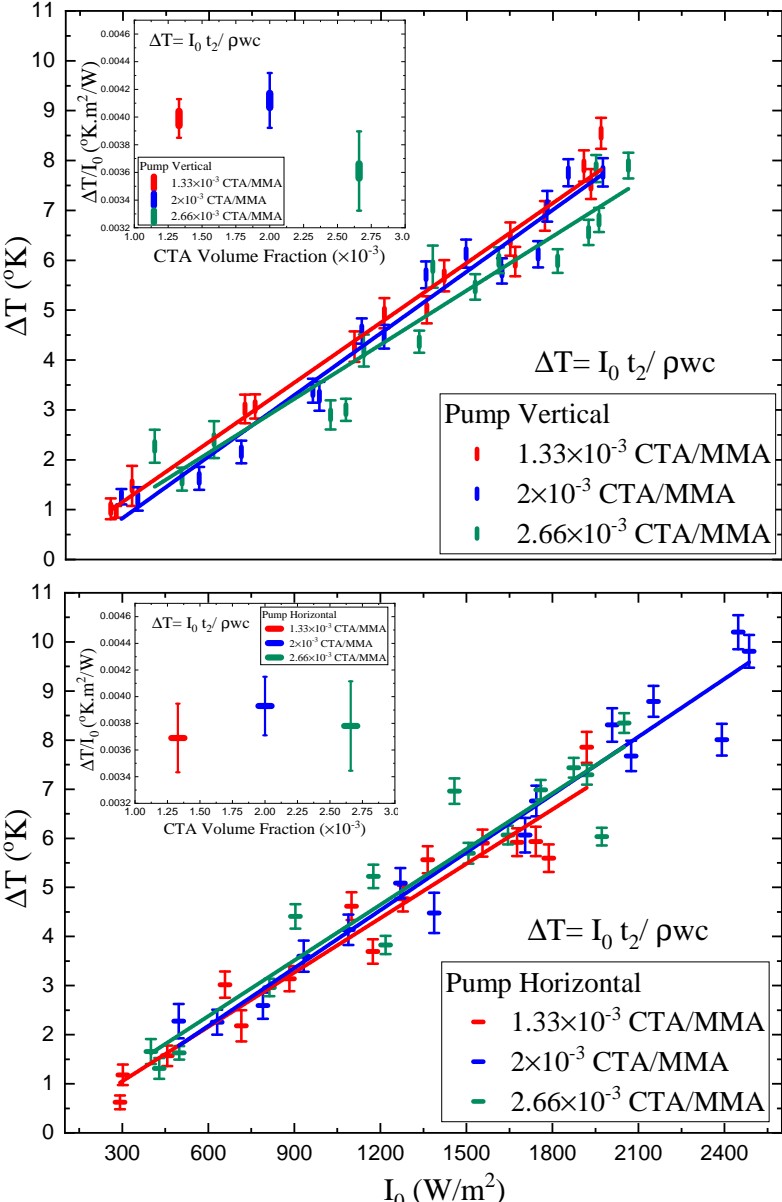

**Figure 16.** Temperature change as a function of intensity calculated from Equation (4) using the time constants measured from the photomechanical response, the density, heat capacity, and the sample's dimensions. The insets show the temperature change per unit of absorbed intensity as a function of CTA volume fraction averaged over all intensities. The plots use $t_2$ to get $\Delta T$ for vertically and horizontally polarized pumps.

Figure 17 summarizes all the results obtained in this paper by providing plots of the intensity-dependent linear photomechanical response $\kappa_\sigma^{(1)}(I_0)$, defined by Equation (36) as a function of intensity. Included are plots of the theory of the heating contribution that use the temperature change determined from Equation (3) in the heating model given by Equation (7), which draws from the parameters shown in Table 2. These parameters are determined from fits to the temperature-dependent stress shown in Figure 9 to the function given by Equation (6). The response along the two principle axes of the fiber sample are denoted by "|" and "—" for the pump polarization along and perpendicular to the long axis (mounted vertically) for three polymer chain lengths as controlled by the amount of chain transfer agent used during polymerization. Additionally included are plots of the experimentally determined values for the same quantities, which are represented by smooth lines that are determined from fits to the data. The dashed curves are the

heating and molecular reorientational contributions determined from the polarization-dependent measurements.

The solid curves, which show the fit functions to the measured values, and the dashed curves, which show the heating contribution determined from the polarization-dependent measurements, fall on top of each other. The polarization-dependent measurement shows that the degree of molecular reorientation is negligible. Aside from the polymer prepared with $2.66 \times 10^{-3}$ CTA volume per MMA volume, the pure heating model is consistent with the theoretical results. However, the trend of the heating model's prediction and the data are opposite to each other, but on average, they agree. This is not surprising, given that the model has no adjustable parameters and relies only on the temperature-dependent stress measurement and time constant measurement, both of which may be biased by systematic errors.

We have found the temperature-dependent stress measurement of the polymer prepared with $2.66 \times 10^{-3}$ CTA volume per MMA volume to vary significantly between measurements, perhaps because of the high sensitivity of the material's properties to temperature variations due to its lower glass transition temperature. This lack of reproducibility for the heating model in this polymer is consistent with the heating theory being off by a factor of two, from the experimentally determined values.

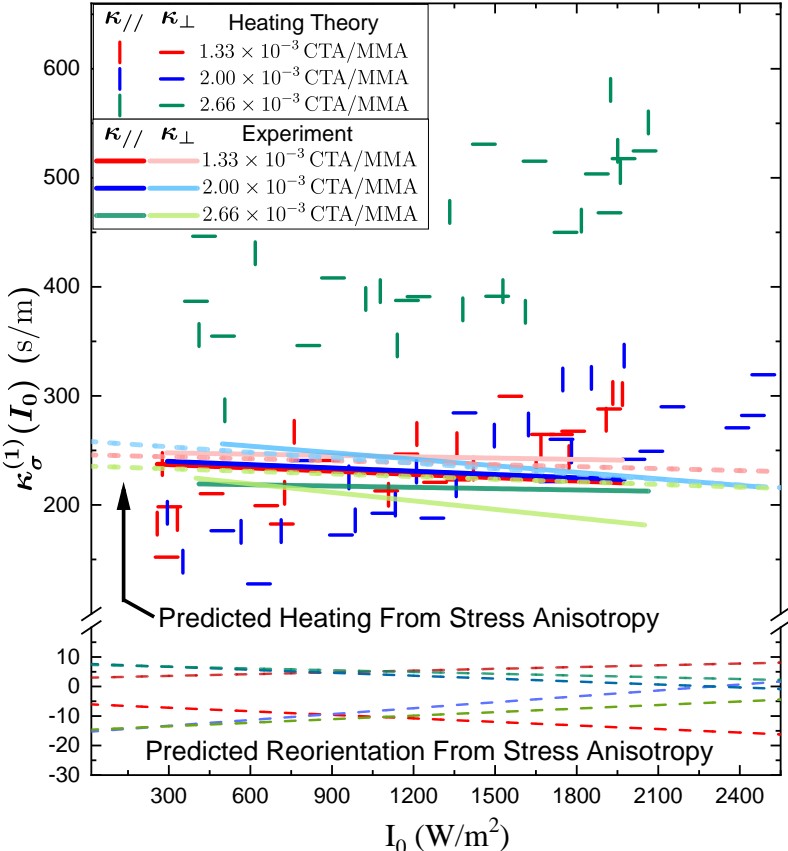

**Figure 17.** The vertical and horizontal lines (representing the pump polarization relative to the vertical) show the predicted heating contribution to the photomechanical response as a function of intensity using the theory (Equation (7)) and the parameters determined from the fits in Figure 9 of the temperature-induced stress. The smooth lines show the experimentally determined intensity-dependent photomechanical constant determined by the fit of the data to a quadratic function in the intensity, as shown in Figure 11, and identical to the results shown in Figure 12, but, here the points are connected to make smooth curves. $\kappa^{(heat)}(I)$, $\kappa^{(re)}_{//}(I)$, and $\kappa^{(re)}_{\perp}(I)$ are also plotted and determined from the tensor properties of the polarization-depended response using Equations (19)–(21).

The figure of merit is typically defined by the linear photomechanical response and Young's modulus $E$ to be given by $\left(\kappa_\sigma^{(1)}\right)^2/E$. The intensity-dependent figure of merit is a generalization of the figure of merit for an illuminated sample, providing another degree of freedom that can be controlled. It depends on the intensity-dependent photomechanical response $\kappa^{(1)}(I_0)$ and the intensity-dependent Young's modulus $E(I_0)$, which are determined using Equation (32), where $T(I_0)$ is the slope of the plots in Figures 15 and 16 at intensity $I_0$, yielding

$$E = \frac{E_0}{\left(1 + \left(\frac{a+bI_0}{T_c}\right)^n\right)},\tag{38}$$

where $a$ and $b$ are the intercept and slope of the graphs in Figures 15 and 16. So

$$FOM(I_0) = \frac{(\kappa^{(1)}(I_0))^2}{E(I_0)} = \frac{(\kappa^{(1)} + 2\kappa^{(2)}I_0)^2}{\frac{E_0}{(1+(\frac{a+bI_0}{T_c})^n)}},\tag{39}$$

Figure 18 shows a plot of the intensity-dependent figure of merit as a function of intensity as determined from fits to the data to get the appropriate parameters. The intensity-dependent figure of merit is observed to decrease as a function of intensity. While the decrease in Young's modulus with increased intensity works to improve the figure of merit, the decrease in the intensity-dependent photomechanical response wins out, resulting in a decreased figure of merit. As such, the best figure of merit is for a dark sample.

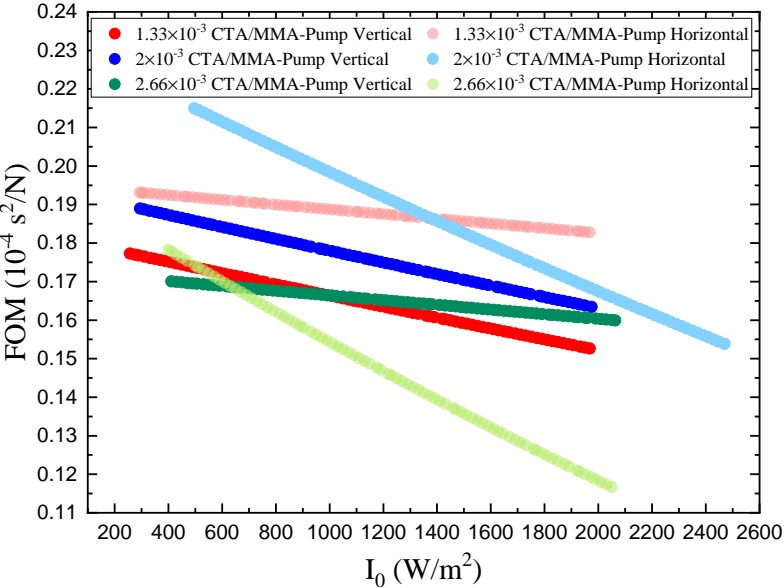

**Figure 18.** Figure of merit as a function of intensity for fibers with different CTA concentrations and measured for two orthogonal polarizations. The result is derived from the intensity-dependant photomechanical response $\kappa^{(1)}(I_0)$ and the intensity-dependent Young's modulus $E(I_0)$, given by Equation (39).

Finally, we determine the temperature-dependent figure of merit due to the heating mechanism for the three polymers to determine if heating the material might improve it. Using the semi-empirical heating model of the photomechanical response given by Equation (7) and a fit to the measured temperature-dependent Young's modulus given by

Equation (32), the temperature dependence of the figure of merit is determined to be of the form

$$FOM(T) = \frac{(\kappa^{(1)}(T))^2}{E(T)} = \frac{\left(\frac{n(\sigma-\sigma_0)}{T\left(1+\left(\frac{T_0}{T}\right)^n\right)} \cdot \frac{t_1}{\rho cw}\right)^2}{\frac{E_0}{(1+(\frac{T}{T_c})^n)}}.$$

(40)

Figure 19 shows the temperature dependence of the figure of merit. For all chain lengths and light polarizations, it increases with increased temperature by a factor of about three when the temperature is raised by 14 K. Thus, adjusting the operational temperature might be one avenue for improving the photo-mechanical response of a material.

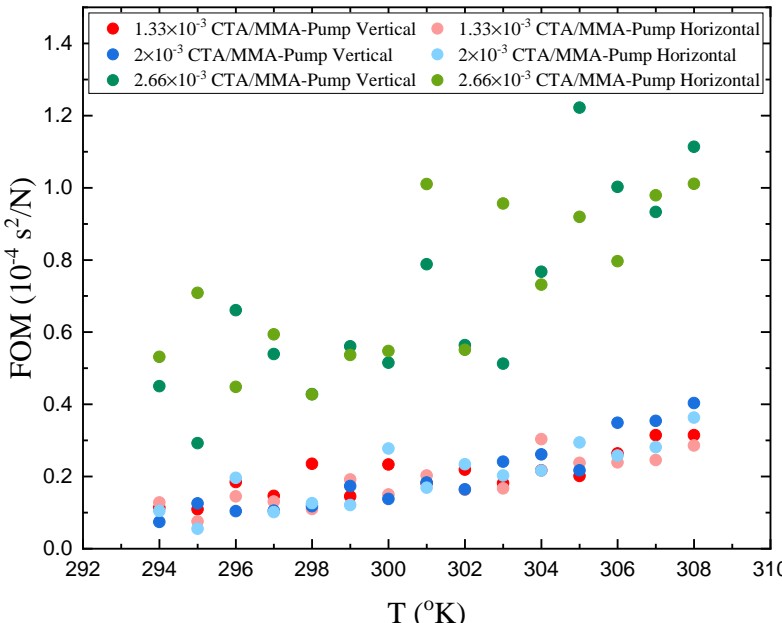

**Figure 19.** Figure of merit as a function of temperature for fibers with different CTA concentrations. The result is derived using $\kappa^{(1)}(T)$ and $E(T)$ as shown in Equation (40).

### 4.2.2. Separating Photothermal Heating and Angular Hole Burning Using the Tensor Response

$R^{(re)}(I_0)$, the ratio of the orientational response to the heating response for light polarized along the axis between the clamps, is given by Equation (14), which is expressed in terms of the intensity-dependent photomechanical constants $\kappa_{//}^{(1)}(I_0)$ and $\kappa_{\perp}^{(1)}(I_0)$. Equation (36) relates the intensity-dependent photomechanical constants to the linear and quadratic photomechanical constants. These, in turn, are the fit parameters determined from a plot of the photomechanical stress response as a function of intensity, and determined for the polymers studied here from the data in Figure 11. The zero-intensity limit, $R^{(re)}(0)$, is the value we observe in most experiments since the sample is typically not illuminated before the pump light is turned on to actuate a photomechanical response in the material.

Table 6 summarizes the zero-intensity results for the DR1-doped PMMA fibers of varying CTA volume fraction. The fractional contribution to molecular reorientation $R^{(re)}(0)$ is small and consistent with zero within experimental uncertainty, showing that heating is the dominant mechanism. $\kappa^{(heat)}$ is positive, indicating that thermal expansion elongates the fiber sample along its long axis. The magnitudes of $\kappa_{//}^{(re)}(I)$ and $\kappa_{\perp}^{(re)}(I)$ are also within experimental uncertainty of vanishing, aside from the medium-chain-length sample.

The negative sign of $\kappa_{//}^{(re)}(I)$ in all measurements, particularly in the medium-chain-length polymer, is marginally statistically significant. The results are consistent with molecular reorientation, which decreases the length of the samples along the polarization axis of the light, and the positive sign of $\kappa_{\perp}^{(re)}(I)$ shows an increase in the samples' length along its long axis when excited by light perpendicular to it. The fact that all of the measured reorientational contributions are consistently of the correct sign suggests that the uncertainties determined from fits to the data might be overestimated.

**Table 6.** Contribution of photothermal heating and angular hole burning to the photomechanical response of DR1/PMMA fibers pumped with 488-nm light for different CTA volume fractions.

| CTA Volume Fraction | $\kappa_{//}^{(1)}(I)/\kappa_{\perp}^{(1)}(I)$ | $R^{(re)}(0)$ | $\kappa_{//}^{(heat)}(I) = \kappa_{\perp}^{(heat)}(I)\,(s/m)$ | $\kappa_{//}^{(re)}(I)\,(s/m)$ | $\kappa_{\perp}^{(re)}(I)\,(s/m)$ |
|---|---|---|---|---|---|
| $1.33 \times 10^{-3}$ | $0.96 \pm 0.28$ | $-0.02 \pm 0.19$ | $246 \pm 80$ | $-6 \pm 46$ | $3 \pm 23$ |
| $2 \times 10^{-3}$ | $0.91 \pm 0.08$ | $-0.06 \pm 0.05$ | $259 \pm 22$ | $-15 \pm 14$ | $8 \pm 7$ |
| $2.66 \times 10^{-3}$ | $0.94 \pm 0.23$ | $-0.04 \pm 0.16$ | $231 \pm 49$ | $-10 \pm 36$ | $5 \pm 18$ |

We find that the effect of heating and molecular reorientation are independent of polymer chain length, since all the photomechanical constants are the same within their experimental uncertainties.

Figure 20 shows a plot of the relative reorientational, $R^{(re)}(I_0)$, as a function of background illumination $I_0$. Such experiments are not common, but, as with temperature, they provide the experimenter with an additional degree of freedom that can be varied to provide yet another perspective of the photomechanical response. The ratio $R^{(re)}(I_0)$ is observed to vary significantly over the intensity range measured. The medium-chain-length sample, for example, starts with a ratio of $-0.06$ and crosses zero at the higher intensity range. The short chain polymer (highest CTA concentration) shows the most dramatic change, with the sign changing at low intensity and the ratio increasing by a factor of almost 4.

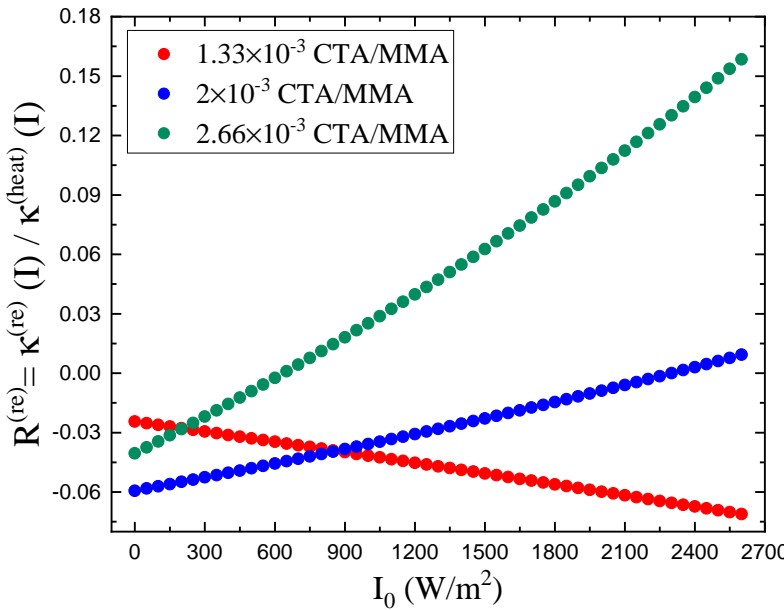

**Figure 20.** A plot of the ratio of molecular reorientation to heating along the sample's axis when pumped with 488-nm light for three different CTA concentrations.

This data shows the most dramatic effect as a function of chain length; the average slope becomes more positive as the chain length decreases. Decreased chain length results in greater mobility, which should lead to a larger reorientational contribution, hence a more

negative value. A possible explanation of the observation is as follows. For long chain lengths, high background irradiation softens the material, making the molecules more mobile and hence increasing the molecular reorientational contribution. For short chain lengths, the molecules are more mobile when softened by the background light. The light also induces molecular reorientation through light-induced torque on the induced dipole moment. Such effects are the topic of future research.

## 5. Conclusions

We have presented an exhaustive study on the photomechanical and thermomechanical response of DR1-doped PMMA fibers of varying CTA volume fraction as a function of temperature, intensity, polarization, and prestress along the two principle axes of the uniaxial material. The data has been analyzed with two models to determine the underlying mechanisms.

The studies start with characterizing the thermo-rheological properties of the DR1-doped PMMA polymer as a function of time, temperature, pre-stress, and polymer chain length. Measurements of the time-dependence of the photomechanical stress in response to a pump laser determine the steady state photomechanical constants, as well as the time constant of the response. The measured thermo-rheological properties and photomechanical time constant are used as input parameters to a model of the heating contribution to the photomechanical response, which predicts its absolute steady state value with no adjustable parameters. The measured results agree with the model's predictions, suggesting that photothermal heating is the dominant mechanism. The maximum temperature increase above ambient induced by the light at the highest intensity is determined to be on the order of $\Delta T \approx 10\,\text{K}$.

The photomechanical experiment determines the stress response to light of a fiber sample that is constrained between a fixed clamp and a force sensor. The force is measured as a function of time, temperature, pump light polarization, and pre-stress. The photomechanical response is found to be well-modeled by a quadratic function of pump intensity and the fit functions are used to determine the effective intensity-dependent linear photomechanical coefficient. The time response is found to follow a single exponential, suggesting that one mechanism is at work. Furthermore, all time constants appear to be independent of intensity and pump light polarization, but the turn-on and turn-off time constants might be slightly different with a small dependence on polymer chain length.

We apply our second independent model of the uniaxial photomechanical stress tensor to photothermal heating, angular hole burning, and molecular reorientation with the strategy of using polarization-dependent pump measurements to separate the mechanisms. Such measurements also show that photothermal heating is the dominant mechanism, with a hint of molecular reorientation at the edge of statistical significance.

Finally, we use the parameters measured here to determine the photomechanical figure or merit (FOM), which is a measure of the conversion efficiency of light to mechanical energy. The FOM is found to be approximately independent of polarization and polymer chain length. We also evaluate the intensity- and temperature-dependent FOM using the intensity- and temperature-dependent photomechanical constant and Young's modulus. We find that the FOM decreases with increased intensity but increases with increased temperature by a factor of two or three, with a temperature increase of 15 K. Thus, soaking a material at elevated temperature might be a method for making existing materials more efficient.

To conclude, we find that the photomechanical effect in dye-doped polymers is dominated by photothermal heating, and that molecular reorientation plays a negligible, though perhaps a statistically significant, role. The approach presented here has been applied to dye-doped polymers but is generally applicable to any material that absorbs light and is made of molecules that change shape or orientation when excited with light. Future studies will focus on more novel materials and sample geometries.

**Author Contributions:** Conceptualization, M.G.K.; methodology, M.G.K.; software, B.Z.; validation, B.Z., Z.G. and M.S.K.; formal analysis, M.G.K. and Z.G.; investigation, Z.G., M.S.K, M.G.K., B.Z. and R.O.; resources, M.G.K.; data curation, B.Z., Z.G. and M.G.K.; writing—original draft preparation, Z.G. and M.G.K.; writing—review and editing, M.G.K., Z.G., B.Z., R.O., M.S.K.; visualization, Z.G. and M.G.K.; supervision, M.G.K.; project administration, M.G.K.; funding acquisition, M.G.K. All authors have read and agreed to the published version of the manuscript.

**Funding:** This research was funded by The National Science Foundation, Directorate for Engineering (ENG) grant number EFRI-ODISSEI:1332271.

**Data Availability Statement:** Data produced by this work will be available on or before 1 March 2022 at https://www.dropbox.com/sh/3yqk1a2c6d98mls/AACiOlOZ0ZHfetebCiqEkq_Ba?dl=0.

**Acknowledgments:** The authors thank the National Science Foundation (EFRI-ODISSEI:1332271) for supporting this work.

**Conflicts of Interest:** The authors declare no conflict of interest.

**Sample Availability:** Samples of the compounds are available from Z.G., M.S.K., or R.O.

## Abbreviations

The following abbreviations are used in this manuscript:

| | |
|---|---|
| DR1 | Disperse Red 1 |
| PMMA | poly (methyl methacrylate) |
| CTA | chain transfer agent |
| POF | polymer optical fiber |
| MMA | methyl methacrylate |
| IPA | isopropyl alcohol |
| n-BMC | n-butyl mercaptan |
| PMDS | pentamethyldisilane |
| FOM | figure of merit |
| $T_g$ | glass transition temperature |

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
