# Peer review of "Photothermal and Reorientational Contributions to the Photomechanical Response of DR1 Azo Dye-Doped PMMA Fibers"

_applsci, doi:10.3390/app12010315_

Round 1
Reviewer 1 Report
The authors present an extensive study of the photothermal and re-orientational contributions to the photomechanical response of ago-dye-doped PMMA. In particular, they focus on the DR1 molecule, which is widely used for producing optically reconfigurable polymer composites.
I found the manuscript to be well organized, although a few minor details may help to improve it. Some statements are provided as known facts but the general reader may need some additional references (or perhaps they are simply misplaced within the paragraph). As an example, the paragraph starting in line 65 begins with three sentences that are presumably supported by the references provided in the remaining of the paragraph. The order of this paragraph must be revised as it is somewhat confusing (e.g., in line 68, ..."They concluded that the best...," who are they?). Some terminology is also introduced without previous definitions: "hole burning" (line 107) is used without any further description. The meaning of this is readily explained in subsequent sections of the manuscript, but its use in the initial sections without a brief description may seem confusing to non-specialized readers. A similar construction is used in the paragraph starting in line 119: the statement "...A temperature increase causes the materials length to change – which in a clamped configuration prevents the material from changing length – resulting in a stress on the force sensor, which is in series with one of the clamps...." includes parts of the experimental setup, but this is not described until later sections.
In section 2.2, the authors mention that a "common measurement configuration is a sample stretched between two clamps..." Further elaboration is needed for this statement: common measurement configuration for what? This is the classical setup for tensile tests and it should be described as such. A simple scheme in this section showing the basic elements of this setup would help to avoid potential confusions to some readers (or perhaps simply refer the reader to Fig. 4 in the manuscript). In the same paragraph, the authors state that this configuration is "ideal to detect processes in which polarized light anisotropically changes the alignment or shape of molecules." Once again, this might be obvious for people working on mechanical characterization of materials but not for the general reader. Why is this an ideal configuration for these purposes? The authors should provide a reason for this statement. In the same section (line 162) the authors also state that "...Both angular hole burning and reorientation are at play, with angular hole burning dominating at early times and re-orientation kicking in at later times." Are the time scales of these processes known? If so, the authors should provide appropiate references. Otherwise, they should ellaborate on the reasoning of this statement.
Regarding the experimental details, I have only one comment. In the last paragraph of section 3.3 the authors state that the power absorbed by the tested samples is "underestimated by a few percent." Compared to the detailed elaboration of the theoretical section, it seems odd to find such a vague explanation of the experimental measurements. What is a few percent? A better estimate should be provided. What is the refractive index of the samples? This can be used to estimate at least the reflectivity.
In summary, other than my comments above, I find the paper to be suitable for publication.
Typos:
Caption Fig. 2 (oriented is misspelled in the last sentence).
Line 320: Argon is wrongly spelled.
Line 451: "and" is repeated.
Caption Fig. 17 (measured is misspelled in the first sentence).
Reviewer 2 Report
This manuscript describes the Photothermal and Re-orientational Contributions To the Photomechanical Response of DR1 Azo-dye-doped PMMA Fibers. The paper contains elements of novelty but needs major revisions. I would recommend the following amendments and questions:
- Please redraft the abstract, the sentence: This work introduces a comprehensive experimental and theoretical study aimed at understanding the photothermal and molecular reorientation contributions to the photomechanical effect of polymers doped with azo dyes, which are known to change conformation through photo-isomerization is too convoluted.
- Please underscore the novelty of your work in the introduction part, please add the most recent literature on this subject.
- In introduction, please add clearly purpose of your work.
- Please explain what is the advantage of your materials over other polymeric materials .
- In experimental part please add information about materials (producer, purity) and methods.
- Please explain DRI abbreviations.
- In whole manuscript is lack of literature references (????)
- Please add polymerization method.
- Please add chemical structures of used monomers in experimental part.
- Propose a mechanism for polymerisation.
- The article is definitely too long, making it very difficult to read. The article should be rewritten, shortened, some of the studies and figures, (19!!!), should be moved to the Supplementary material.
Round 2
Reviewer 2 Report
Accepted in present form.